

# Summertime evaporation over two lakes in the Schirmacher oasis, East Antarctica

Elena Shevnina[1], Timo Vihma[1], Miguel Potes[2,3] and Tuomas Naakka[1]

[1] Finnish Meteorological Institute, Helsinki, Finland
[2] Institute of Earth Sciences (ICT), Institute for Advanced Studies and Research (IIFA), University of Évora, Évora, Portugal
[3] Earth Remote Sensing Laboratory (EaRSLab), Institute for Advanced Studies and Research (IIFA), University of Évora, Évora, Portugal

*Correspondence to*: Elena Shevnina (elena.shevnina@fmi.fi)

**Abstract.** The study quantified uncertainties in the bulk-aerodynamic method and combination formulas being applied in estimations of summertime evaporation over two lakes in the Schirmacher oasis, East Antarctica. The evaporation over the lakes was measured by the eddy-covariance (EC) technique during the austral summers (December – January) in 2017–2018 and 2019–2020. These direct measurements showed that summertime evaporation over two lakes varied from 0.3 to 5.0 mm d$^{-1}$. Depending on the ice cover presence, the average evaporation varied from $1.6 \pm 0.1$ mm d$^{-1}$ in December to $3.0 \pm 0.2$ mm d$^{-1}$ in January – February. In summer, the lakes were warmer than the ambient air, and the largest day-to-day variations in evaporation were associated with variations in the wind speed. The EC measurements were used as a reference for evaluating the uncertainties of the indirect methods. The bulk aerodynamic method gave the most accurate estimates of evaporation over two lakes (of 6 – 8 %), and this method showed acceptable skill scores (by two selected criteria) in estimation of the daily evaporation during the lakes' ice breaking-up and open water periods. This method is recommended for hydrological (lake water balance) applications required for operational (short term) decision making. Most of the combination formulas underestimated the summertime evaporation by 27–73 %.

## 1 Introduction

Antarctica shelters numerous water bodies containerizing meltwater in a complex hydrological system of lakes and streams. Every summer, approximately 60 000 lakes are formed on the surface of the ice shelf and margins of the continental ice sheet. The largest surface lakes are found in East Antarctica, where the interannual variabilities of lake surface area is the highest, and in the Antarctic Peninsula (Stokes et al., 2020; Dirscherl et al., 2021; Shen et al., 2025). These lakes speed-up calving of the marginal glaciers, which contributes to the global sea level rise and increases the risk of collapse of ice shelves due to hydrofracturing (Rignot et al., 2004; Banwell et al., 2013; Arthur et al., 2022). Providing a moisture source to the atmosphere through evaporation, these surface lakes may also affect local weather by increasing precipitation, warming near-surface temperature and fostering fog formation.



There are approximately 115 permanent settlements in Antarctica, most of them located along the continental coast, in rock oases and sub-Antarctic islands (Antarctic station catalog COMNAP, 2025). Since the COVID-pandemic, the number of people visiting Antarctica has increased by 40%, and is expected to grow further by approximately 12.5% per year (Bastmeijer et al., 2023). The nearest lakes often serve freshwater for the settlements, and evaluation of the lakes' water resources (water balance) is a crucial task for their managers (Kaup, 2004; Sokratova, 2011; Dhote el al., 2021). Relying on

short-term weather forecasts, the managers make operational decisions and expectations on lake water inflow/outflow. The decisions on maintenance of settlements and investments rely on seasonal and climate-scale perspectives (Lan et al., 2025). The Antarctic lakes house unique lifeforms (Rothschild and Mancinell, 2004; Andersen et al., 2011; Keskitalo et al., 2013). Understanding the hydrological regime and lakes' water balance is important to save these ecosystems (Faucher et al., 2019). It requires  reliable hydrological observations on lakes, and methods allowing evaluation of components of the water (mass)

balance equation.

The lake water balance equation describes changes in the lake volume depending on changes in the surface inflow, outflow runoff, precipitation and  evaporation/sublimation over the lake surface area, artificial water withdrawal, underground inflow/outflow runoff, etc. Contributions of some of the above-mentioned components to the total lake volume change may be minor, whereas others may be essential, depending on a particular case. This equation is usually applied in the integrated

form, with an integration time period (e.g., a day, season, year or decade) depending on the application (Chebotarev, 1975). In regions with polar climate, evaporation is a key component of the water (mass) balance of the lakes (Le et al., 2016; Leppäranta et al., 2020; Wang et al., 2020; Shi et al., 2024). The evaporation is, however, difficult to measure directly, and, therefore, it is estimated using indirect methods needing only a few meteorological and hydrological observations (Finch and Hall, 2001). The uncertainties of the indirect methods are often estimated relying on the eddy-covariance (EC) technique

(Tanny et al., 2008; Tran et al., 2023; Shi et al., 2024) but is rarely used in Antarctica.

In hydrological applications, the evaporation over lakes is often evaluated using the  empirical relationships between the evaporation, air-water moisture deficit, and wind speed (Keijeman, 1974;  Finch and Hall, 2001) also known as combination formulas. They have been applied for the lakes in Antarctica even though their uncertainties are unknown (Borghini et  al., 2013; Shevnina and Kourzeneva, 2017; Dhote et al., 2021). Many of these combination formulas, however, underestimate

the evaporation over the local lakes by up to 72 %, and new empirical formulas need verification against independent measurements (Shevnina et al., 2022).

The bulk-aerodynamic method is often used to assess evaporation/sublimation over  lakes and glaciers in Antarctica (Clow et al., 1988; Bliss et al., 2011; Leppäranta et al., 2016). In this method, the turbulent exchange (mass-transfer) coefficients account for atmospheric stability, which is calculated through the Monin–Obukhov framework, incorporating empirical

dimensionless gradient functions (Brutsaert, 1982). The empirical gradient functions are site specific, and evaluated from the eddy-covariance (EC) measurements on the lakes (Franz et al., 2018; Ala-Könni et al., 2022; Guseva et al., 2023). According



to Shevnina et al. (2022), in coastal Antarctica, the bulk aerodynamic method underestimated the summertime evaporation over an ice-free lake by over 32 % , and it is not clear how good it is for the lakes during the ice break-up period.

The study aims (a) to quantify the uncertainties of the combination equations and bulk-aerodynamic method, and (b) to
verify the new empirical formulas with the independent observations. This study estimated the evaporation over two lakes in the Schirmacher oasis, East Antarctica using the micrometeorological (eddy-covariance) and hydrological measurements collected in two summers (December – January) in 2017–2018 and 2019–2020.

## 2 Study area

The Schirmacher oasis (70°45′ S, 11°38′ E) is situated approximately 80 km from the coast of Lazarev Sea, Dronning Maud
Land (SA in Fig. 1). The oasis is an ice-free rock outcrop approximately 20 km long and 3 km wide, engolated in the west–northwest to east–north-east direction (Simonov and Fedotov, 1964). The relief is hillrocks with absolute heights up to 228 m above sea level (asl). The rock hills mostly consist of gneiss mixed with basalts (Sengupta, 1991). The oasis shelters two scientific bases operated year round (highlighted in red in Fig. 1), and two camps occupied seasonally (highlighted in blue). The scientific bases have supported long term meteorological observations according to the guidelines of the World
Meteorological Organization (WMO) since the early 1960s. The settlements, ice runways and coastal bases are connected with year-round ice roads, which suffer from the lakes and temporal streams formed on the ice surface in summertime.

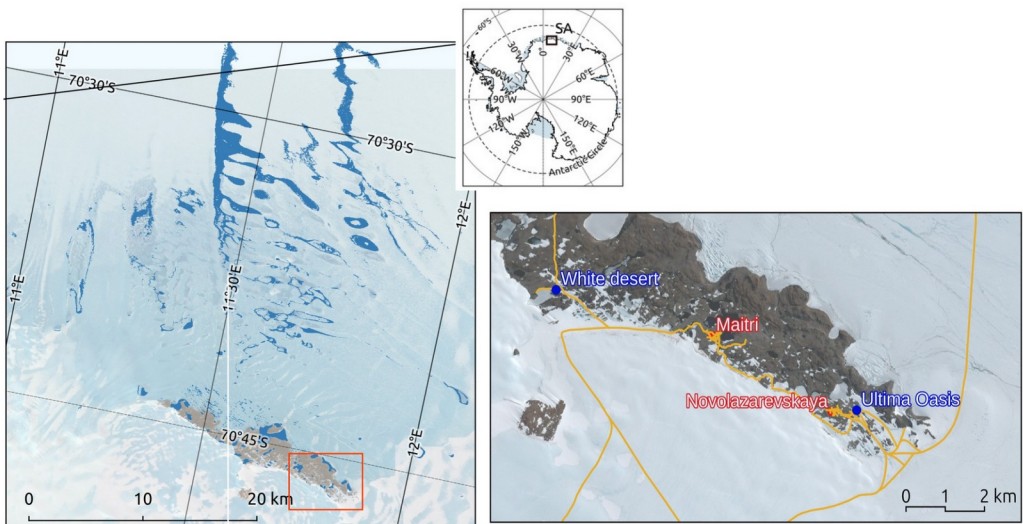

**Fig. 1. Location of the Schirmacher oasis (SA) and social infrastructure: year round (red) and seasonal (blue) settlements connected by roads (yellow lines, © Humanitarian OpenStreetMap Team (HOT), 2020. Distributed under the Open Data Commons Open Database License (ODbL) v1.0.). © Google Maps, 2019.**

Antarctic weather is characterized by gravity-driven katabatic winds that flow down slope from the continental interior towards the coast. The katabatic winds play a key role in the regional climate, affecting sea ice formation, distribution of





atmospheric moisture and precipitation in Antarctica (Nygård et al., 2013; Grazioli et al., 2017; Naakka et al., 2021), and enhancing the sublimation/ablation over the local lakes (Dugan et al., 2013; Leppäranta et al., 2016). The region of the Schirmacher oasis is featured by the katabatic winds (Richter and Bormann, 1995). The maximum wind speed reaches up to 38 ms$^{-1}$ in the austral winter (June – August) when the air temperature ranges from −4.5 to −12.9 °C (Govil et. al., 2016). During the austral summer (December – February), the average wind speed is 8.0 ms$^{-1}$, and the average accumulated

precipitation is less than 10 mm water equivalent, falling as snow (Asthana et al., 2019).

More than 300 surface lakes in the Schirmacher oasis are documented by the SCAR Antarctic Digital Database (Gerrish et al., 2020). Most of these lakes are freshwater and free of ice for 6 – 12 weeks in the summer, when the lakes are mixed down to the bottom because of strong winds. The largest lakes are connected by ephemeral streams (Sokratova, 2011; Phartiyal et al., 2011). In winter, winds remove snow cover from the lake ice. This study focuses on the evaporation over two lakes

differing in their surface area and depth. Lake Zub/ Priyadarshini is a water body with a maximum depth of 6.0 m (mean depth is 2.9 m), and a surface area of 35000 m$^2$ (Dhote et al., 2021). The lake is free of the ice during 6–8 weeks each summer, and its freshwater is used to support the needs of the year-round scientific base Maitry (Khare et al., 2008). In 2017 – 2018 and 2019 – 2020, Lake Zub/Priyadarshini was ice free from the beginning of January to the end of February (6–7 weeks). Lake Glubokoe is of a maximum depth of 34.5 m (mean of depth is 13.1 m) and the surface area of 147000 m$^2$

(Loopman et al., 1988). The lake is normally ice-covered year round (Kaup, 2005), but in recent years the lake has been ice free almost every summer (Sharov and Tolstikov, 2020). In February 2018 and 2020 the lake was ice free for 2 – 3 weeks.

**3 Data and methods**

**3.1 Micrometeorological and hydrological observations**

The air temperature, barometric pressure, wind speed/direction and water vapour concentration were measured on a tower

equipped with the EC open-path system IRGASON by Campbell Scientific. The EC measurements were collected during 38 days (1, January to 8, February, 2018) in the period when Lake Zub/Priyadarshini was free of ice (Shevnina et al. 2022). This study analyzed the observations collected during the experiment on Lake Glubokoe. The lake water temperature measurements cover the period from 8 December, 2019 to 12 February, 2020. The EC system was operated from 7 December 2019 to 8 January 2020, yielding an observation period of 33 days. The IRGASON was deployed on a mast

installed at a distance of 10 m landward of the shore of Lake Glubokoe. It was placed at a height of 1.8 m and its water vapor sensor was directed toward 144° south-eastwards (Fig. 2).





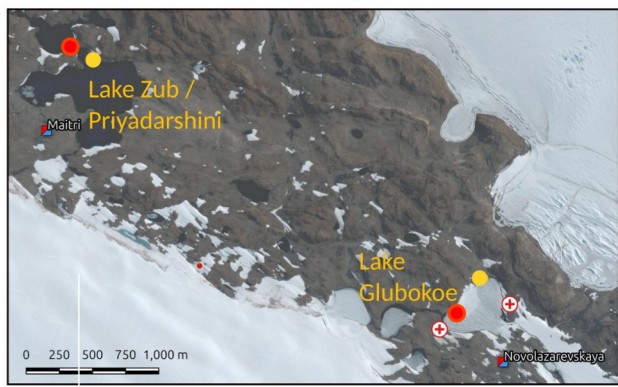
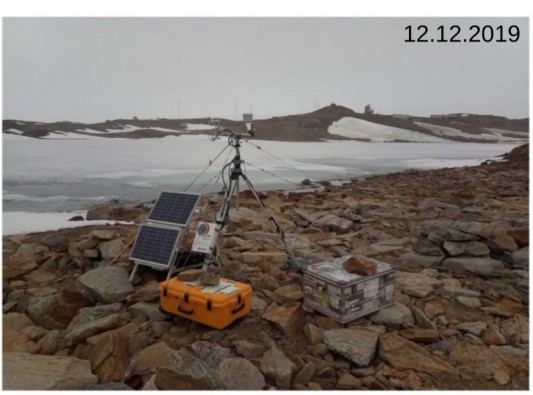

a                                                b

**Figure 2. Location of the instrumentation on the shores of two lakes (a): HOBO logger (red dots), the EC system (yellow dots), and digital cameras collecting the panoramic images (rounded red crosses), © Google Maps 2019. The photo on (b) shows the EC system installed on the shore of Lake Glubokoe.**

The lake surface water temperature (LSWT) was measured by the HOBO instrumentation (Fig. 2a), which logged the temperature every 30 minutes with the accuracy ±0.44 degree. The HOBO logger was installed at a depth of 0.02 m. Shevnina et al. (2022) give detailed information on the instrumentation used in the experiments. On Lake Glubokoe, the experiment covers the ice break-up on the lake. The fraction of ice over the lake surface was evaluated from the panoramic images taken by a digital camera (red crosses in Fig. 2a).

The wind speed, atmospheric pressure, air temperature and water vapor concentration measured by the EC system, and the LSWT measured by the HOBO loggers were used (a) to calculate the evaporation applying the indirect methods, (b) to define an empirical parametrization for the transfer coefficient of moisture in the bulk-aerodynamic method, and (c) to verify the empirical parameters with independent data. Also, the observations by the EC system were used to calculate the air relative humidity following Hoeltgebaum et al. (2020) and saturation vapor pressure (specific humidity) corresponding to the lake surface temperature and air (Stull, 2017).

### 3.2 Methods

The EC technique (Burba, 2013) was used to measure evaporation from the two lakes. The raw measurements were post-processed following Potes et al. (2017). The data were processed to remove spikes (Vickers and Mahrt, 1997), and the spikes detected were counted for quality control. The EC instrumentation collects the gas concentration in 0–360° wind directions, but the only data collected from a sector associated with the footprint over the lake was accounted for. The measurements were filtered by the footprints defined according to Kljun et al. (2004). It is assumed that the EC method gives the most





accurate estimates for the evaporation with the instrumental error not exceeding 7 % for our EC system (Shevnina et al., 2022).

The indirect methods to assess the evaporation included in common hydrological practices (as the term in lake water balance) are often based on the combination formulas, and in this study evaluated the evaporation over Lake Glubokoe using the combination formulas suggested by Penman (1948) given here following Tanny et al. (2008), Doorenbos and Pruitt (1975), Odrova (1979), Shuttleworth (1993) and Shevnina et al. (2022).

In the bulk-aerodynamic method, evaporation is defined as the vertical surface flux of water vapor due to atmospheric turbulent transport, and is calculated on the basis of the difference in specific humidity between the surface (ice or water) and the air (Brutsaert, 1985). In our study, the evaporation was calculated as follows:

$$E = \rho C_{Ez} w_z (q_s - q_{az}) \tag{1}$$

where E is the evaporation (kg m$^{-2}$ s$^{-1}$); $\rho$ is the air density (kg m$^3$), $C_{ez}$ is the turbulent transfer coefficient for moisture, $q_s$ is the saturation specific humidity corresponding to the lake surface temperature (kg kg$^{-1}$), $q_{az}$ is the air specific humidity (kg kg$^{-1}$), and $w_z$ is the wind speed (ms$^{-1}$). The subscript z refers to the observation height. The transfer coefficient of moisture was calculated following the parametrization schemes suggested by Heikinheimo et al. (1999), Andreas (1986), Arya (1988) and Fedorovich et al. (1991) and wind-dependent relationships acquitted from the EC measurements on lakes. The schemes were verified with independent data. The values by Heikinheimo et al. (1999) were given for $z = 3$ meters, and converted to our observation height of 1.8 meters using Launiainen and Vihma (1990).

The evaporation estimates applying the bulk-aerodynamic method and combination formulas were compared to the estimate based on the EC technique (considered as a reference). To evaluate the skill of the indirect methods, the root mean square error ($RMSE = \sqrt{\sum_{1}^{n} \left( E_{EC} - E_{mod} \right)^2}$) was used following Moriasi et al. (2007). We also applied the s/σ (SSC) criteria to define if a method is acceptable to be used in hydrological operational (short-term) practice or not, and we followed the recommendation that the SSC < 0.8 (Popov, 1979). In the SSC criteria: $s = \sqrt{\sum_{i=1}^{n} \left( E_{EC}^i - E_{mod}^i \right)^2 / (n - m)}$ and where, $E_{EC}$ is $\sigma = \sqrt{\sum_{i=1}^{n} \left( E_{EC}^i - \bar{E}_{EC} \right)^2 / n}$ the evaporation by the eddy covariance method, $E_{mod}$ is evaporation by an indirect method; $\bar{E}$ is the mean evaporation, (mm d$^{-1}$); $n$ is the length of the series (33), and $m$ is the number of empirical coefficients in the relationships (equal to 2). The mean daily evaporation over the observational period and its error ($\sigma_{\bar{E}} = \dfrac{\sigma}{\sqrt{n}}$) were calculated following (Rozhdestvenskiy and Chebotarev, 1974).



## 4 Results

### 4.1 Weather conditions, lake surface water temperature and lake ice cover

During the period from 7 of December 2019 to 8 January 2020, the daily air temperatures ranged from –4.9 ℃ to 5.1 ℃ taken on 1.1 ℃ average. The daily relative humidity was 56 % on average, and it varied from 42 to 80 %, and there were two days when the relative humidity was higher than 70 % (marked vertical black line on Fig. 3, top), and they are 25 December 2019 and 3 January 2020. These days, winds were lower than 4.9 ms$^{-1}$ (average for the experiment) and most of them were coming from the northern directions. The wind speed ranged from 0.1 to 13.0 ms$^{-1}$, and the strongest winds were observed on 9 December 2019 and 1 January 2020 (Fig. 3, bottom). Fig. 3 shows the relative humidity, water vapour concentration, air temperature and atmospheric pressure observed by the EC system during the experiment on Lake Glubokoe.

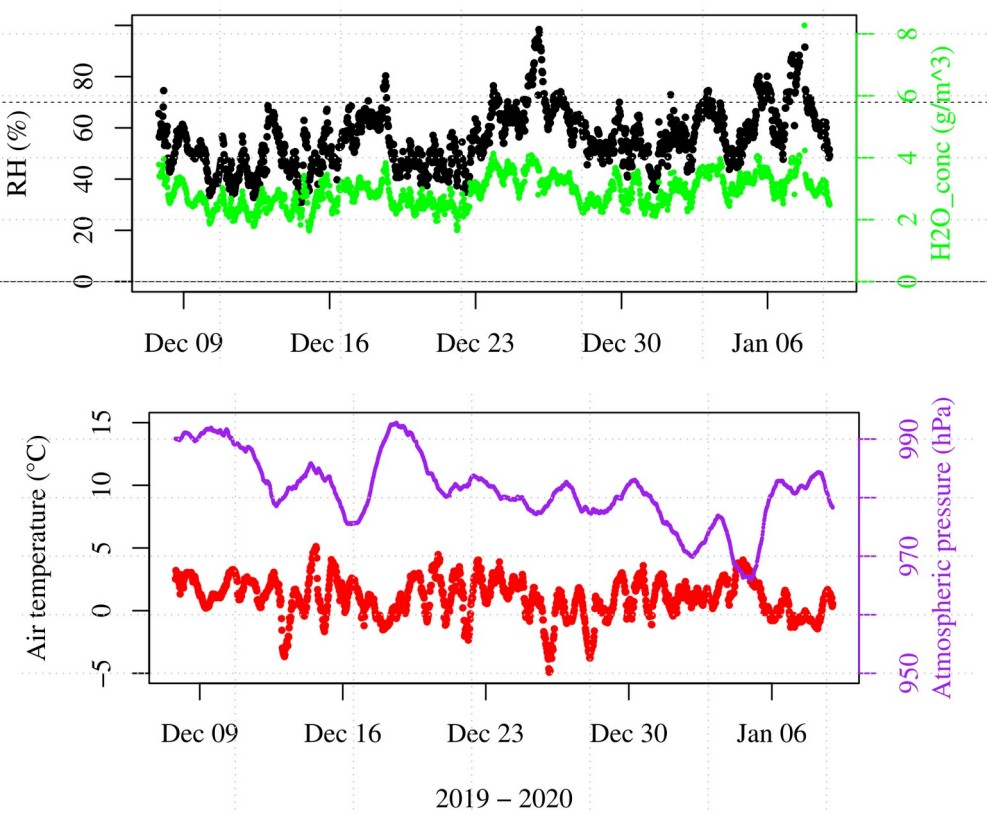

**Figure 3. The relative humidity (green), water vapour concentration (blue), air temperature (red) and atmospheric pressure (purple) observed from 7 December 2019 to 8 January 2020.**




During the field experiments, the lakes were warmer than the ambient air on most days. In January 2018, the LSWT in Lake
Zub/Priyadarshini was 3.9 ºC, which was on average 4.7 ºC higher than the air temperature (Fig. 4a). The difference between
the LSWT and air temperature varied from –0.5 ºC (2–3 January, 2018, during the storm) to 10 ºC (25–26 January, 2018) for
Lake Zub/Priyadarshini. The average daily LSWT of Lake Glubokoe was 3.1 ºC, and it ranged from 0.6 to 5.3 ºC during 7
December 2019 to 15 February 2020. In this period, the lake was 2.4 ºC warmer than the air on average (Fig. 4b). The largest
difference in the LWST and air temperature was observed on 25 December 2019 (marked by vertical green line on Fig. 4b)
when the relative humidity was the highest.

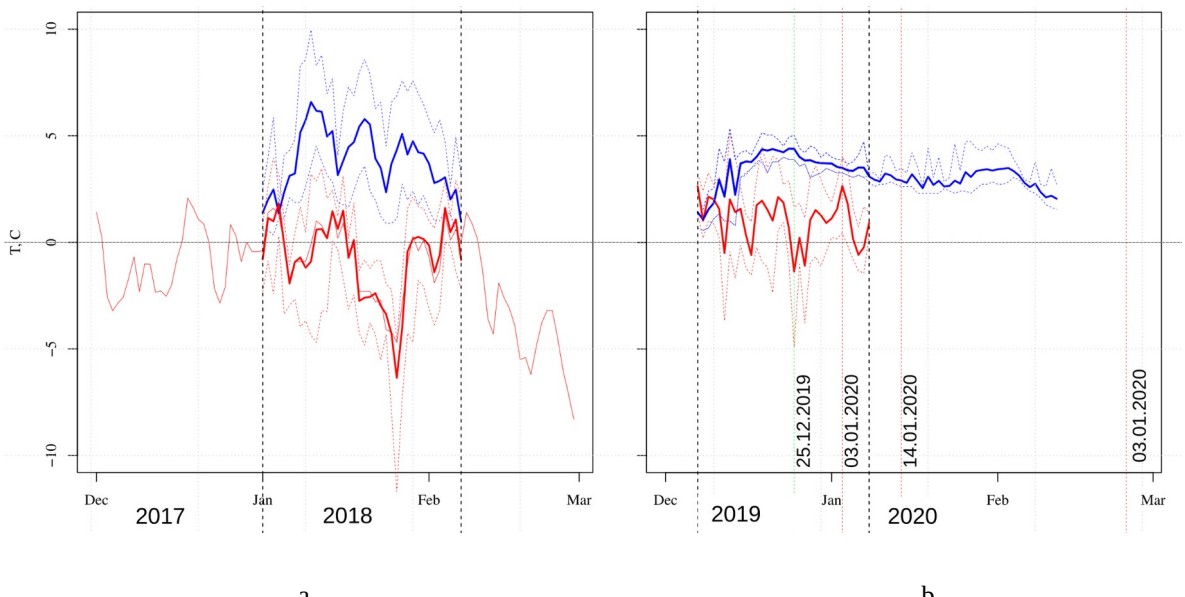

a                                                                b

**Figure 4. The daily minimum, average and maximum for the air temperature (red lines) and the LSWT (blue lines) measured on
the shore of Lake Zub/Priyadarshini (a) and Lake Glubokoe (b). The pink line shows the air temperature according to the
observations at Maitri site (a).**

The ice cover in Lake Glubokoe was documented in a series of digital images taken with two cameras (marked by the red
crosses in Fig. 2a) from mid-December 2019 to the end of February 2020. The fraction of lake ice was evaluated from the
images. The ice breaking-up period on Lake Glubokoe lasted until the end of January, and the ice free period lasted for
approximately 2.5 weeks in February, 2020. The EC measurements were done in the period of 33 days when the ice melted
from 30–35 % of the lake surface. Figure 5 shows the images of the lake ice cover which were taken 3 January, 14 January
and 25 February, 2020 (red vertical lines in Fig. 4).





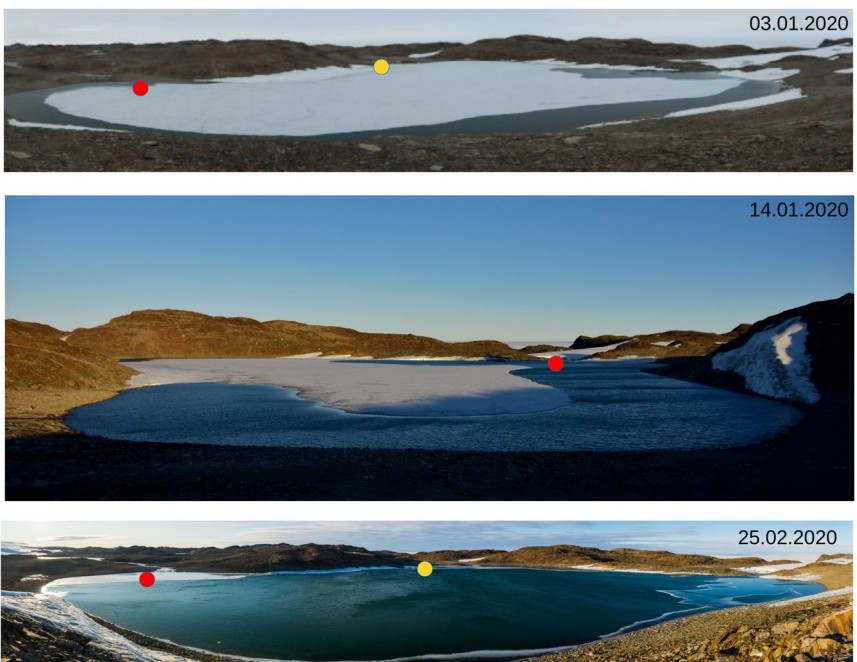

**Figure 5. The panoramic images of Lake Glubokoe and location of the instrumentation (the red and yellow dots show the HOBO logger and the EC system). Photo by Dmitrii Emelyanov.**


### 4.2 Evaporation, the EC technique

The evaporation over Lake Glubokoe was estimated from the EC measurements processed for each 30 minute period during 33 days (7 December 2019 – 8 January 2020). The EC system was installed on the lake shore in a location allowing it to cover the sector between 90 and 225 degrees (Fig. 6a) and to collect the measurements from the lake surface. The height of the EC system is given over 90 % of the footprint within the lake surface (X90, in metres) which varies between 50 and 150 m in the experiment on Lake Glubokoe (Fig. 6 b).






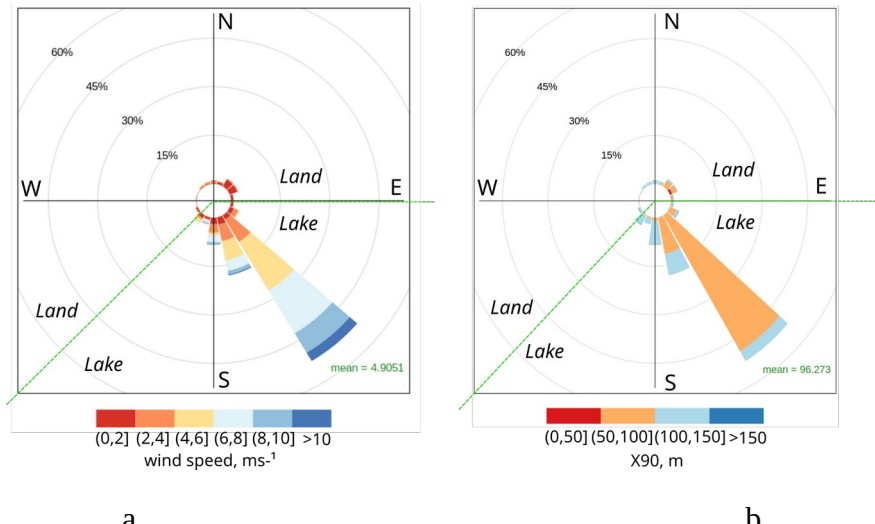

**Figure 6. The wind roses (a) and length of the footprint (b) for the experiment on Lake Glubokoe. The green lines outline the sector of the wind directions covering the lake surface.**

The raw 30-min measurements were filtered by (a) the sensors' signal strengths, (b) number of gaps in the observations, and

(c) by the footprint or sector of wind directions covering the lake (green lines in Fig. 6 and red lines in Fig. 7, top). The percentage of the filtered measurements did not exceed 20 % of total data (red dots in Fig. 7, top); most of them were collected on days with low winds coming from directions outside the footprint (Fig. 7, bottom).





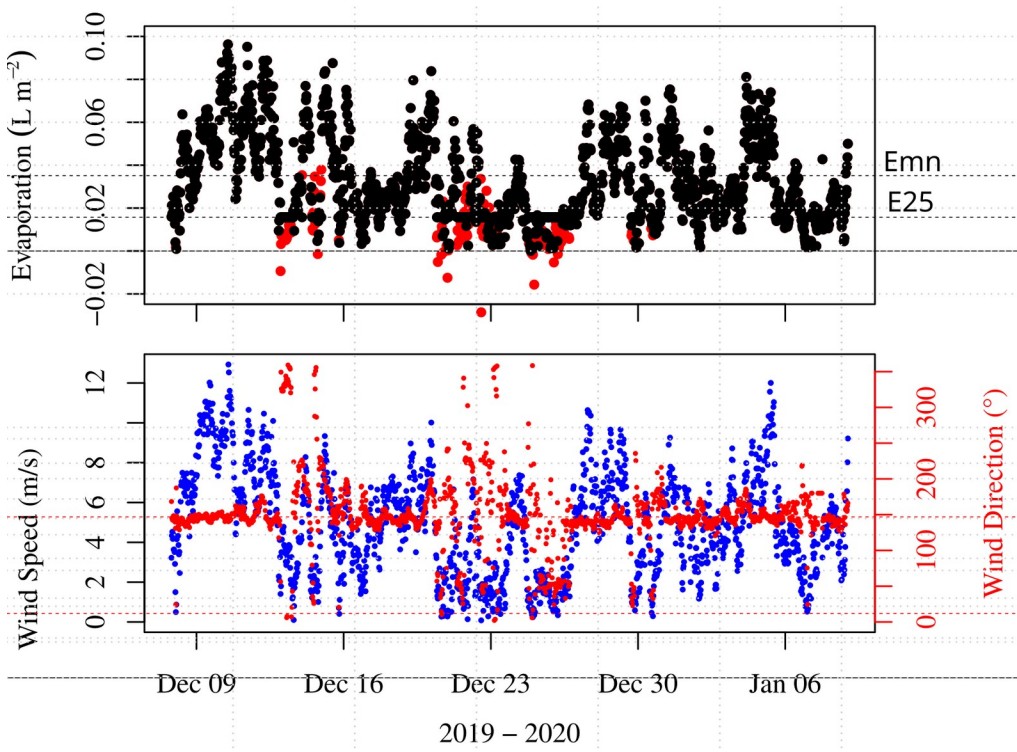

**Figure 7. The 30 min time series of the evaporation (top) and wind speed and wind direction (bottom) measured by the EC system.**
**On the top, the red dots indicated the measurements collected outside the footprint outlined by the red lines (bottom).**

In this study, these gaps were replaced by 0.25 quantile (E25, L m$^{-2}$) and average of evaporation (Emn, L m$^{-2}$) estimated for the period from 7 December 2019 to 8 January 2020 (33 days). After the filling gaps, the evaporation over each day was calculated from 30 minutes measurements. Depending on the value applied to fill the gaps (E25 or Emn), the sum of evaporation over Lake Glubokoe was 50 mm or 54 mm for the period of 33 days. The daily evaporation was 1.5 ± 0.1 mm d$^{-}$

$^{1}$ and 1.6 ± 0.1 mm d$^{-1}$ on average if the gaps were replaced by E25 and Emn, respectively. For the reference estimates, the daily evaporation was calculated by fitting the gaps by the average evaporation. The daily evaporation varied between 0.3 and 3.2 mm d$^{-1}$, and the largest evaporation was observed during the days with the strongest winds (13 – 15 December, 2019; 3 – 4 January, 2020).

Figure 8 shows the diurnal cycle of the air temperature (top) and LSWT (bottom) for two lakes: Lake Zub/Priyadarshini (a)

and Lake Glubokoe (b). Daytime air temperatures were warmer than nighttime, with differences of 5 ºC in January 2018 (top, a) and of 2.5 ºC in December 2019 (top, b). At Lake Zub/Priyadarshini during the ice free stage, the sub-daily cycle of lake surface water temperature (LSWT) was distinct (bottom, a), with nighttime minimums around 2.5 ºC (23:00–02:00) and daytime maximums exceeding 6.0 ºC (12:00–14:00). At Lake Glubokoe during the ice break-up stage, the LSWT showed weaker diurnal variation, remaining close to 3.5 ºC (Fig. 8 bottom, b).





**Figure 8. The diurnal cycles of air temperature (top) and lake surface water temperature (bottom) according to the measurements on Lake Zub/Priyadarshini (a) and Lake Glubokoe (b).**

Figure 9 shows the diurnal cycles of the wind speed (top), the saturation pressure deficit (middle) and evaporation (bottom) which were estimated from the observations on Lake Zub/Priyadarshini (a) and for Lake Glubokoe (b). In both cases, the highest wind speed was observed at nighttime (03:00 – 04:00 AM), and it was lower at daytime (04:00 – 05:00 PM). For the lake in the ice free stage (Lake Zub/Priyadarshini), the evaporation is largest during the daytime hours (11:00 AM – 01:00 PM), and its diurnal cycle is likely follows to the cycle of the saturation vapor pressure deficit reflecting the difference between the lake surface temperature and air temperature.





a                                                    b

 

**Figure 9. The diurnal cycle of wind speed (ms⁻¹, top), saturation vapor pressure deficit (kPa, middle) and evaporation (mm h⁻¹, bottom) estimated from the observations on Lake Zub/Priyadarshini (a) and Lake Glubokoe (b).**

For the lake in the ice break-up stage (Lake Glubokoe), the diurnal cycle of evaporation showed peaks in the early morning hours (06:00–8:00 AM) and lowest values in the late evening hours (09:00– 11:00 PM). It closely followed the wind speed cycle, with both peaking at night and decreasing during the day (Fig. 9b, top). This pattern reflects the complex interplay of factors (air-water temperature gradient, air humidity, wind speed, solar radiation) that vary throughout the day. The early morning peak may be linked to the overnight cooling of air, which increases the temperature difference between the lake surface and air. Analogously, the evening minimum is probably due to the smallest temperature difference.

### 4.3 Evaporation, uncertainties of the indirect methods

We estimated the evaporation over Lake Glubokoe on the basis of five combination formulas given by Shuttleworth (1993), Doorenbos and Pruitt (1975), Penman (1948) and Odrova (1979). Following these combination equations, the average evaporation varied between $0.4 \pm 0.1$ mm d⁻¹ and $1.1\pm 0.1$ mm d⁻¹ (Table 1), and it was 27 % – 73 % less than the reference (EC) method, which yielded an average evaporation of $1.6\pm 0.1$ mm d⁻¹. The formula by Shevnina et al. (2022) has been developed using the data collected on Lake Zub/Priyadarshini (2017 – 2018), and with the independent observations on Lake Glubokoe it gave only 6 % underestimation with respect to the evaporation measured by the EC technique. The SSC of all combination formulas is higher than 0.80, and the RMSE varies between 1.3 and 1.6 mm d⁻¹.

**Table 1. The evaporation over Lake Glubokoe on the basis of five combination formulas ($E_{CE}$, mm day⁻¹) and their skill scores.**

| Author(s) (year) | $E_{CE}$, mm d⁻¹ | | Sum mm p⁻¹ | $E_{EC} / E_{CE}$ | skill scores | |
| --- | --- | --- | --- | --- | --- | --- |
| | Average | Max | | | RMSE | SSC |
| Penman (1948) | $0.6 \pm 0.1$ | 1.6 | 20 | 2.7 | 1.3 | 2.1 |
| Doorenbos and Pruitt (1975) | $0.8 \pm 0.1$ | 2.4 | 28 | 1.9 | 1.5 | 2.4 |
| Odrova (1979) | $0.4 \pm 0.1$ | 1.1 | 13 | 4.1 | 1.3 | 2.2 |
| Shuttleworth (1993) | $1.1 \pm 0.1$ | 2.6 | 37 | 1.4 | 1.3 | 2.1 |
| Shevnina et al. (2022) | **$1.5 \pm 0.2$** | 5.0 | 50 | 1.1 | 1.6 | 2.6 |

We estimated the evaporation on the basis of the bulk-aerodynamic method with different parametrization applied for the transfer coefficient of moisture. We calculated the transfer coefficient for the height of the instruments (different in the two experiments), and for the height of 10 meters ($C_{EN10}$) to compare our results with others. The wind-dependent transfer coefficient was derived from the EC measurements in Zub/Priyadarshini (2017 – 2018), and then applied to calculate the evaporation over Lake Glubokoe. We suggest the following parametrization:



$C_{E10}= 0.0000119\ w_2 + 0.00114\ (w_2 < 13\ \text{m}^{-1})$ and $C_{E10}= 0.0013\ (\text{if}\ w_2 >= 13\ \text{ms}^{-1})$     (2)

$C_{E2}= 0.0000193\ w_2 + 0.00153\ (\text{if}\ w_2 < 13\ \text{m}^{-1})$ and $C_{E2}= 0.0018\ (\text{if}\ w_2 >= 13\ \text{m}^{-1})$     (3)

Figure 10 shows the values of the transfer coefficients estimated for the standard height of 10 m.

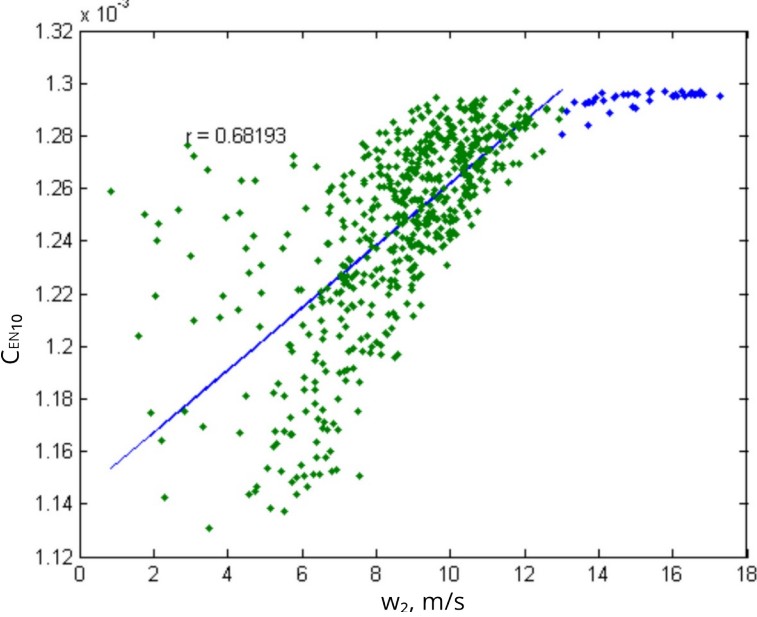

**Figure 10. Dependence of the 10-m neutral transfer coefficient for moisture ($C_{EN10}$) on the 2-m wind speed ($w_2$, ms$^{-1}$) over Lake Zub/Priyadarshini.**

We also calculated the transfer coefficient of moisture on the basis of the measurements on Lake Glubokoe following Andreas (1986), Arya (1988) and Fedorovich et al. (1991). Then, we used the independent measurements on Lake Zub/Priyadarshini to verify the calculated values. The values of $C_{EN10}$ varied between $1.46\cdot10^3$ and $2.10\cdot10^3$ for the lakes in the Schirmacher oasis (Table 2).

**Table 2. Transfer coefficients for moisture ($C_{ENz}$) in the bulk-aerodynamic method. Notations: z refers to the height; the underlined values were recommended to apply in estimation of the evaporation over two selected lakes.**

| Parameterization scheme | $C_{ENz}$ , z = 10m | Lake Zub / Priyadarshini z = 2 m | Lake Glubokoe z = 1.8 m |
|---|---|---|---|
| Heikinheimo et al., 1999 | 0.001246 | 0.001011 | 0.000995 |
| Wind-dependent coefficient | Eq. (2) | Eq. (3) | |
| Arya (1988) | 0.001459 | 0.001184 | 0.001166 |
| Andreas (1986) | 0.002099 | 0.001702 | 0.001676 |

Table 3 shows the daily lake evaporation we estimated with four parameterizations for the transfer coefficient of moisture following Heikinheimo et al. (1999), wind-dependent (this study), Andreas (1986), Arya (1988) and Fedorovich et al.



(1991). Following these parametrizations, the average daily evaporation varied between $2.0 \pm 0.1$ mm d$^{-1}$ and $3.0 \pm 0.2$ mm d$^{-1}$ in case of the Lake Zub/Priyadarshini. The parametrization by Andreas (1986) gave the best estimate for the evaporation being $3.0 \pm 0.2$ mm d$^{-1}$ on average. In the case of Lake Glubokoe, the best estimate for the evaporation being $1.5 \pm 0.1$ mm d$^{-1}$ (on average) was obtained with the transfer coefficient after Arya (1988) while the reference (EC) method assessed the evaporation as $1.6 \pm 0.1$ mm d$^{-1}$ (ie. 6 – 8 % less than the indirect method). Depending on the parametrization of the transfer coefficient, the bulk-aerodynamic method either overestimated the evaporation up to 38 % or underestimated it up to 20 % (Table 3).

**Table 3. The evaporation over two lakes ($E_{BA}$ mm d$^{-1}$) estimated after the bulk-aerodynamic method with four schemes of parametrization for the transfer coefficient of moisture ($C_{ENz}$).**

| Parameterization scheme | Lake Zub / Priyadarshini | | | | Lake Glubokoe | | | |
|---|---|---|---|---|---|---|---|---|
| | Average | Max | Sum | $E_{EC} / E_{BA}$ | Average | Max | Sum | $E_{EC} / E_{BA}$ |
| Heikinheimo et al. (1999) | $2.0 \pm 0.1$ | 3.5 | 75 | 1.6 | $1.3 \pm 0.1$ | 2.3 | 42 | 1.3 |
| Wind-dependent coefficient | $2.6 \pm 0.2$ | 4.8 | 100 | 1.2 | $1.9 \pm 0.2$ | 3.6 | 63 | 0.9 |
| Arya (1988) | $2.1 \pm 0.2$ | 3.8 | 79 | 1.5 | $\underline{1.5 \pm 0.1}$ | 2.7 | 50 | 1.1 |
| Andreas (1986) | $\underline{3.0 \pm 0.2}$ | 5.4 | 114 | 1.0 | $2.2 \pm 0.2$ | 3.9 | 71 | 0.8 |

Figure 11 shows the daily evaporation estimated by the direct EC (x-axis) against those estimated by the bulk-aerodynamic method using three parameterizations for the transfer coefficient of moisture which was calculated from the independent observations in each case. In this figure, HK refers to Heikinheimo et al. (1999), AN to Andreas (1986), AF to Arya (1988), and WD is the wind-dependent coefficient derived from the EC measurements collected on Lake Zub/Priyadarshini. Then, the observations on Lake Glubokoe were applied to verify the calculated values from the independent measurements. In the case of Lake Zub/Priyadarshini, the transfer coefficient of moisture was calculated from the EC measurement on Lake Glubokoe.





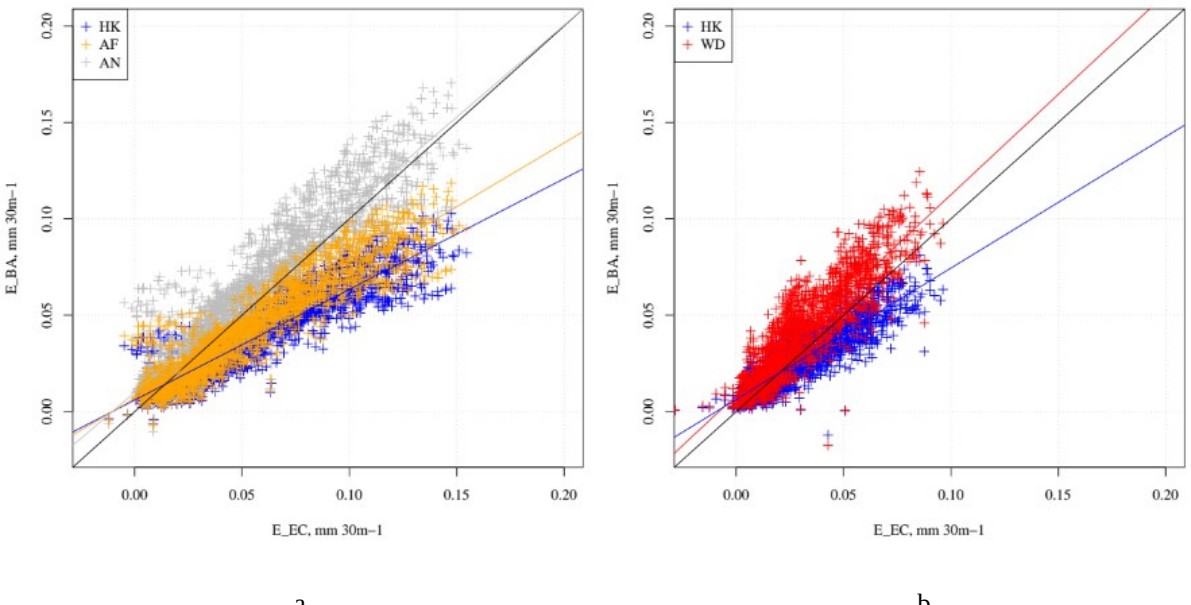

a                                        b

**Figure 11. The measured (x-axis) evaporation and simulated (y-axis) with the bulk-aerodynamic method applying different parameterizations for the transfer coefficient of moisture: (a) Lake Zub/Priyadarshini and (b) Lake Glubokoe. HK refers to Heikinheimo et al. (1999), AN to Andreas (1986), AF to Arya (1988), and WD is the wind-dependent coefficient derived from the measurements collected in 2017 – 2018.**

275    Table 4 shows the root mean square error (RMSE) depending on the parametrization (model) applied for the transfer of moisture in the bulk-aerodynamic method. It varied from 0. 4 to 0.8 mm d$^{-1}$, the SSC criteria were less than  0.8 in both cases.

**Table 4. The skill scores of the bulk-aerodynamic method using various parameterizations for the transfer coefficient of moisture.**

| Parameterization schemes | Lake Zub / Priyadarshini | | Lake Glubokoe | |
|---|---|---|---|---|
| | RMSE | SSg | RMSE | SSC |
| Heikinheimo et al., 1999 | 1.2 | 1.2 | 0.5 | 0.8 |
| Wind-dependent transfer coefficient | 0.7 | 0.7 | 0.5 | 0.9 |
| Arya (1988) | 1.1 | 1.2 | **0.4** | 0.7 |
| Andreas (1986) | **0.7** | 0.7 | 0.8 | 1.2 |

Among the indirect methods, the bulk-aerodynamic method (with specific parametrization for the transfer coefficient of

280    moisture) showed the best fit to the direct measurements of evaporation over two lakes in the Schirmacher oasis. Two





35

parametrization schemes (namely Andreas (1986) and Arya (1988)) gave acceptable skill scores (SSC < 0.8) in simulation of daily evaporation over the lakes.

**5 Discussion**

This study yielded estimates of the uncertainties of the combination formulas and bulk-aerodynamic method, and it was

based on the eddy-covariance measurements collected on two lakes in East Antarctica. The field experiments were done in December – February 2017 – 2018 and 2019 – 2020 covering the ice-free stage on Lake Zub/Priyadarshini and the break-up stage on Lake Glubokoe. The evaporation over Lake Glubokoe during the break-up period varied between 0.3 and 3.2 mm d$^{-1}$, and it took 1.6 ± 0.1 mm d$^{-1}$ on average. It resulted in 54 mm evaporation in the period of 33 days (7 December 2019 – 8 January 2020). It is almost twice less than the evaporation over the ice-free Lake Zub/Priyadarshini estimated as 3.0 ± 0.2

mm d$^{-1}$ on average for the period lasting from 1 January to 7 February 2018 (Shevnina et al., 2022).

The sub-daily cycle of the evaporation closely followed the wind speed's cycle with peaks at nighttime (11:00 PM –02:00 AM) and lows at daytime (12:00 AM– 02:00 PM); and was weaker in December (ice break-up stage) than in January (ice free stage). The evaporation increased with wind speed with the maximum up to 5.0 mm d-1 observed during the wind storms. The highest evaporation was measured 2 – 3 January 2018 on Lake Lake Zub/Priyadarshini (ice free stage) and 13 –

15 December 2019 on Lake Glubokoe (ice break-up stage). The EC observations on lake evaporation are very limited in Antarctica; ones, however, can be found for the regions with similar climate (polar deserts). Increasing the lake evaporation with the wind speed has been found for the high-latitude Lake Qinghai, Qinghai-Tibet Plateau (36°N, 3194 m asl) based on the long term EC measurements (Li et al., 2016; Shi et al., 2024). Applying 2 years of observations, Li et al. (2016) found that the evaporation over the ice-free lake reaches up to 12 mm d$^{-1}$ during wind storms, and days with wind speed stronger

than 4 ms$^{-1}$ contribute up 22 % to annual lake evaporation. Lately, Shi et al. (2024) confirmed that the evaporation over the ice free lake is controlled by wind based on the EC observations collected in 2014 – 2019. The authors suggested that the ice sublimation takes 23 % of annual evaporation over Lake Qinghai.

Leppäranta et al. (2016) observed an average summertime average sublimation of 0.7 mm d$^{-1}$ from ice covered and ice free lakes in the Vestfjella mountains, Droning Maud Land (73°S). Further, Leppäranta et al. (2016) observed an extreme event

of summertime lake ice thinning on a small pond on top of a nunatak. During the event wind speeds exceeded 30 m s$^{-1}$ and there was no surface melt. Leppäranta et al. (2016) accordingly attributed the ice thinning to sublimation of 10 – 15 mm d$^{-1}$. No EC observations were available from the site but we checked that according to the bulk-aerodynamic method these values are indeed possible under such a wind speed, if the ice surface temperature is close to 0 °C and the air is very dry. The lake ice ablation (sublimation) over three ice-covered lakes in Tailor Valley, East Antarctica (77°S) is estimated to be 5–31

mm d$^{-1}$ (Dugan et al., 2013), but did not have means to distinguish between the contributions of surface melt and sublimation





to ablation during summer. For winter, Dugan et al. (2013) found sublimation values ranging from 0.2 to 0.7 mm d$^{-1}$. These values for ice-covered lakes in winter are naturally much smaller than our results for mostly ice-free lakes in summer.

The mass and energy balance method, the bulk aerodynamic method, combination formulas, the equilibrium temperature method, empirical formulas, pans and tracers (water isotopes) are among the indirect methods often used to estimate lake

evaporation (Abtew and Melesse, 2013; Bellagamba et al., 2021). Each method has its strengths and weaknesses allowing application in various geographical locations (Finch and Calver, 2008), and we did not find any indirect method suggested for lakes in Antarctica. The evaporation over lakes is often evaluated with the combination formulas, and we found five combination formulas that have been applied over the lakes in Antarctica. These are named after Penman (1948), Doorenbos and Pruitt (1975), Odrova (1979), Shuttleworth (1993) and Shevnina et al. (2022).

Our results showed that Penman's formula (1947) underestimated the lake evaporation by 57 – 60 %. The formulas by Shuttleworth (1993) and Doorenbos and Pruitt (1975) underestimated the daily evaporation by 27 % by 47 % on average, respectively. The formula by Odrova (1979) underestimated the daily evaporation by over 72 – 73 %. This formula has been used to estimate the summertime evaporation over 12 lakes in the Larsemann Hills oasis (69 °S, East Antarctica) and King George Island (69 °S, Antarctic Peninsula), where evaporation varies between 0.8 and 1.7 mm d$^{-1}$ depending on LSWT and

the presence of lake ice cover (Shevnina and Kourzeneva, 2017). In the case of Lake Glubokoe, this formula yielded an average evaporation of 0.4 mm d$^{-1}$, which is less than for the lakes in the Larsemann Hills oasis (LH). This difference can be explained by the fact that the observations on Lake Glubokoe cover the early stage of the lake ice break-up period in December, while the observations on the lakes in the LH were done in January – February when over 60–80% of the lake surface was ice free. The formula by Shevnina et al. (2022) underestimated the daily evaporation by 6 % on average

compared with direct EC observations. For all combination formulas, the methods' skill scores were low (SSg >> 0.80 and RMSE was 1.4 mm d$^{-1}$ on average).

The bulk-aerodynamic method is often used to estimate evaporation; this method is, however, sensitive to a function defining the transfer coefficient of moisture. It is typically considered almost constant with a value of approximately $1.1 \cdot 10^{-3}$ over the ocean (Kantha and Clayson, 2000) and varying between 1.1 and $2.4 \cdot 10^{-3}$ over the lakes and reservoirs (Hicks, 1972;

Guseva et al. 2023). The function (parameterization) defining the transfer coefficient of moisture is site specific, but does not explicitly depend on the lake surface area and depth (Guseva et al., 2023). The bulk-aerodynamic method using the parametrization derived based on observations in a boreal lake underestimated the evaporation over Lake Zub/Priyadarshini by over 30 % (Shevnina et al., 2022). Our results showed that the transfer coefficient of moisture derived according to Arya (1988) or Andreas (1986) allowed reduction of the uncertainties of the bulk-aerodynamic method, which underestimated the

evaporation by only 6 – 8 % in these cases. With the transfer coefficient of moisture varying between $1.4 \cdot 10^{-3}$ and $2.1 \cdot 10^{-3}$, this method demonstrated the following skill scores: SSg < 0.80 and average RMSE of 0.6 mm d$^{-1}$.





The measured (EC) evaporation includes the contributions of both interfacial evaporation from the lake surface and evaporation from spray droplets in the air. It is difficult to distinguish between these contributions but, based on Antarctic observations by Guest (2021), we may assume that under very strong winds the spray evaporation is some 18 % of the
interfacial evaporation. Accordingly, if one wants to apply the wind dependent transfer coefficient (Eqs 3, 4) solely for the interfacial evaporation, the contribution of spray evaporation has to be subtracted from the results. We did not identify clear stability dependence of the moisture transfer coefficient. This was because during the period of the experiments, the lakes were warmer than the ambient air (on average by 4.7 C in January – February 2018). Hence, when the wind was strong, the stratification was close to neutral and the transfer coefficient was large and not sensitive to the stratification.

Shevnina and Kourzeneva (2017) showed that the method used in calculation of the evaporation is important for lakes whose volume changes very little over a summer, and Lake Zub/Priyadarshini is one of such lakes. Its water balance equation includes the precipitation, inlet river runoff (as inflow/positive components) and outlet river runoff, evaporation and water withdrawal (as outflow/negative components). The lake volume is 1032500 $m^3$, and in January-February 2018 it decreased by 40.3 $m^3$, and the discrepancies of the water balance equation was 670.6 $m^3$ (Dhote et al., 2021). The authors calculated the
lake evaporation of 58.5 $m^3$ using the combination formula by Odrova, (1979), which underestimated the evaporation for 72 % according to our results. In absolute values it corresponds to 42.1 $m^3$ (58.5 multiplied by 0.72), which represents approximately 6 % of the discrepancy of the water balance equation. It can be reduced by using a better indirect method while calculating the evaporation. For a better method we suggested the bulk-aerodynamic method with the transfer coefficient of moisture ($C_{ENz}$ , z = 10 m) equaling to $1.4 \cdot 10^{-3}$. The volume of precipitation over the lake was 5.3 $m^3$ in
January-February 2018, which is a negligible component in the lake water balance equation, only representing 0.05% of the lake volume (Dhote et al., 2021). The rest of discrepancy was most probably connected to the methods used to evaluate the inlet/outlet river runoff.

## 6 Conclusions

We quantified uncertainties in the bulk-aerodynamic method and combination formulas applied in estimations of
summertime evaporation over two lakes in coastal Antarctica. The direct measurements by the EC method showed that summertime evaporation over the selected lakes varied from 0.3 to 5.0 mm $d^{-1}$. Depending on the presence of lake ice, the average evaporation varied from 1.6 ± 0.2 mm $d^{-1}$ in the early stage of lake ice break-up (December) to 3.0 ± 0.3 mm $d^{-1}$ for the lake ice-free period (January – February). During the austral summer, the lakes are warmer than the ambient air on most days. The changes in evaporation were associated with changes in wind speed rather than changes in difference between the
near-surface air temperature and lake surface water temperature.

The bulk aerodynamic method yielded the most accurate estimates of evaporation (biases of 6 – 8 % for the mean values) when the moisture transfer coefficient ($C_{ENz}$ , z = 10m) was set to $1.4 \cdot 10^{-3}$ and $2.1 \cdot 10^{-3}$ for Lake Zub/Priyadarshini and Lake



Glubokoe, respectively. This method showed an acceptable skill (SSg < 0.80) in estimation of the daily evaporation over both lakes in the Schirmacher oasis during their ice break-up and open water periods. Hence, the method can be

recommended for hydrological (lake water balance) applications required for operational (short term) decision making.

The selected combination formulas underestimated the daily evaporation over the lakes in the Schirmacher oasis by 27–73 %. With the correction, however, they can be applied to estimate the cumulative summertime evaporation, which is needed for the hydrological applications required for decisions on sites' maintenance and investment. These indirect methods need the measurements of lake surface water temperature.

**Code availability.** The code is available at the supplement. The code for the figures 3, 4, 6, 7, 8 and 9 was written using suggestions given by the AI Github Copilot.

**Data availability.** The dataset is available at https://doi.org/10.5281/zenodo.14823402.

**Interactive computing environment.** Pycharm at Ubuntu (Noble Nombat).

**Supplements.** The evaporation estimated after the EC (direct) and indirect methods are given in ECH_ZB/GL.csv and

BA_daily_ZB/GL.csv.

**Author contribution.** ES designed and ran the field experiments as well as calculated the evaporation applying the combination equations, their uncertainties, and skill scores. TV estimated the evaporation with the bulk-aerodynamic method. MP designed and analyzed the EC measurements in the field experiments. TN analyzed synoptic situation and weather during the experiments. All authors contributed to writing of the manuscript.

**Competing interests.** The contact author has declared that none of the authors has any competing interests.

**Disclaimer.** Publisher's note: Copernicus Publications remains neutral with regard to jurisdictional claims in published maps and institutional affiliations.

**Acknowledgments.** This research has been supported by the European Union's Horizon 2020 research and innovation framework program under Grant Agreement no. 101003590 (PolarRES project). We thank participants of the Annual

Meeting  European meteorological society, EMS (1-7 September 2024, Barselona, Spain) and 7th workshop on parametrization of lakes in numerical weather prediction and climate modeling (20-22 November 2024, Milan, Italy) for questions and discussions during the conferences. We thank Dmitrii Emelyanov, Ivan Kolesnikov and Pankaj Ramji Dhote who were involved in the instrumentation maintenance during the field campaigns.

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
