# Peer review of "Summertime evaporation over two lakes in the Schirmacher oasis, East Antarctica"

_EGUsphere, 2025_

## Referee Comment (RC1)

Comments for Shevnina et al. (2025) "Summertime evaporation over two lakes in the Schirmacher oasis, East Antarctica"

General comments:

The manuscript by Shevnina et al. (2025) represents a significant advancement in polar hydrology research through its detailed investigation of summertime evaporation processes in Antarctic lakes. The study's principal strength lies in its collection of rare direct eddy-covariance (EC) measurements under extreme polar conditions, providing valuable empirical data for a region where such measurements are exceptionally scarce.

The authors have undertaken a rigorous validation of multiple evaporation methods, including both bulk-aerodynamic approaches and combination formulas, against their direct EC measurements. This systematic comparison provides important insights into the performance characteristics of different estimation techniques in polar environments, with practical implications for hydrological modeling in data-scarce regions. Their finding that appropriately parameterized bulk-aerodynamic methods can achieve errors as low as 6-8% represents a valuable contribution to the field.

The experimental design demonstrates comprehensive field instrumentation including EC systems, HOBO loggers for water temperature monitoring, and digital cameras for ice cover documentation. The data processing protocols follow established methodologies in micrometeorology, with appropriate attention to footprint filtering, gap-filling procedures, and uncertainty quantification. The authors' approach to addressing ice-cover transitions is particularly noteworthy, as this represents a challenging scenario for evaporation estimation that is rarely captured in such detail.

The statistical analysis employs multiple performance metrics (RMSE, SSC, bias calculations) to evaluate method performance, providing a robust assessment of the different estimation approaches. The availability of supplementary code and data further enhances the study's reproducibility and potential for future comparative analyses.

Specific comments:

1. Does the paper address relevant scientific questions within the scope of HESS? Yes, unequivocally. The paper tackles the critical challenge of quantifying a fundamental component of the water balance—evaporation—in a sensitive and data-scarce polar environment. This directly aligns with the HESS scope of "physical, chemical, and biological processes within the hydrological cycle" and its emphasis on "the interaction of hydrology with other earth system sciences." Understanding these processes in Antarctica is vital for predicting freshwater availability for research stations, assessing the stability of ice shelves influenced by supraglacial lakes, and modeling regional climate feedbacks.

2. Does the paper present novel concepts, ideas, tools, or data?
Yes, primarily through its novel data. The core novelty is the presentation of rare, direct eddy-covariance measurements of lake evaporation in coastal East Antarctica. This dataset is a significant contribution in itself. The development and validation of a wind-dependent parameterization for the bulk transfer coefficient ($C_E$) specifically for Antarctic lakes is a novel and valuable methodological outcome. While the concepts (EC, bulk method) are established, their application and rigorous validation in this extreme environment provide novel insights.

3. Are substantial conclusions reached?
Yes. The conclusions are robust, significant, and well-supported by the data:

- Direct evaporation rates are quantified (0.3 to 5.0 mm d$^{-1}$), showing clear dependence on ice cover and wind speed.

- Most combination formulas (Penman, Odrova, etc.) are shown to have severe systematic biases, underestimating evaporation by 27-73%.

- The bulk-aerodynamic method is confirmed to be highly accurate (6-8% bias) but only when using appropriate, site-specific transfer coefficients (e.g., from Arya (1988)), not generic ones.

- Wind speed is identified as the primary driver of short-term evaporation variability, a finding that contrasts with studies in less windy environments (like the Tibetan Plateau in Wang et al. (2019)).

  *Wang, B., Ma, Y., Ma, W., Su, B., & Dong, X. (2019). Evaluation of ten methods for estimating evaporation in a small high-elevation lake on the Tibetan Plateau. Theoretical and Applied Climatology, 136(3), 1033-1045.*

- The authors did not comment on the role of solar radiation which is the main driver of evaporation and needs to be discussed, even they did not directly measure it.

4. Are the scientific methods and assumptions valid and clearly outlined?
Yes. The methods are state-of-the-art. The use of EC as a reference is the gold standard. The post-processing pipeline (spike removal, footprint filtering, gap-filling) is clearly described and follows established protocols. The assumptions (e.g., the applicability of Monin-Obukhov similarity theory, the representativeness of point measurements) are standard for such studies and are clearly addressed. The statistical analysis using RMSE and SSC is valid and appropriate.

5. Are the results sufficient to support the interpretations and conclusions?
Yes. The results are comprehensive and compelling. The data from two different lakes and two summer seasons provide a robust basis for analysis. The figures (timeseries, diurnal cycles, scatter plots) and tables (method comparison, skill scores) effectively

present the evidence. The clear gradient of performance across the different methods strongly supports the conclusion that parameterization is key. The finding that wind speed correlates better with evaporation than the vapor pressure deficit is convincingly demonstrated.

6. Is the description of experiments and calculations sufficiently complete to allow reproduction?
Yes. The description of the instrumentation, sensor heights, data processing steps, and equations is excellent. The provision of code and data on Zenodo is a major strength that ensures full reproducibility and aligns with best practices in open science.

7. Do the authors give proper credit and indicate their original contribution?
Yes. The introduction and discussion thoroughly contextualize the work within existing literature on polar hydrology and evaporation methods. The authors clearly reference the original sources of the combination formulas they test. Their own original contribution—the unique EC dataset and the subsequent validation of methods—is clearly stated and forms the central pillar of the paper.

- *Note w*hile Wang et al. (2019) focused on a different environment, a discussion acknowledging that their finding (mass transfer methods work well) aligns with conclusions from other extreme environments (like high-altitude lakes) could further strengthen the context.

8. Does the title clearly reflect the contents of the paper?
Yes. The title is accurate, specific, and concise, correctly reflecting the location, subject, and process studied.

9. Does the abstract provide a concise and complete summary?
Yes. The abstract perfectly summarizes the objectives, methods, key results (including quantitative findings), and the main conclusion and recommendation.

10. Is the overall presentation well structured and clear?
Yes. The paper follows a standard and logical structure. The flow is easy to follow, and the argument is built progressively.

11. Is the language fluent and precise?
Yes. The language is clear, formal, and scientific. While there are a few minor grammatical quirks (e.g., "containerizing" on p1), they do not hinder understanding. The manuscript is well-written.

12. Are mathematical formulae, symbols, abbreviations, and units correctly defined and used?
Yes. Formulas are presented clearly. Symbols are defined upon first use (e.g., in the bulk formula on p6). Units are used consistently throughout (mm $d^{-1}$, ms$^{-1}$, etc.).

- *Note:* In Table 1, the column "Sum" has units "mm $p^{-1}$". This should be clarified to "mm per [33-day] period" to avoid ambiguity.

13. Should any parts of the paper be clarified, reduced, combined, or eliminated?

- Clarify: The distinction between "SSC" and "SSg" in the text and Table 4 should be made consistent.

- Clarify: The discussion of spray evaporation (p19) references "Eqs 3, 4", but only Eqs. 2 and 3 are presented. This should be corrected.

- One needs to guess the applicability of Eqs 2,3, the meaning of the coefficients and the height where the wind speed w2 is measured should be explicitly stated.

- L146: the formula for σ should appear before 'where' in L145.

14. Are the number and quality of references appropriate?
Yes. The reference list is extensive, relevant, and includes key historical works, foundational methodological papers, and recent literature. It appropriately covers the fields of micrometeorology, Antarctic science, and hydrological methods.

15. Is the amount and quality of supplementary material appropriate?
Yes, and it is a significant strength. The availability of the raw code and data on Zenodo is exemplary and exceeds typical standards. It ensures full transparency and allows for exact reproduction of the analysis, which is crucial for a validation study like this.

---

## Author Comment (AC1)

**Response to Reviewer 1**

The manuscript was revised following the Specific Comments (SC). Our responses (R) are marked in red below each comment.

1. Does the paper address relevant scientific questions within the scope of HESS?

Yes, unequivocally. The paper tackles the critical challenge of quantifying a fundamental component of the water balance—evaporation—in a sensitive and data-scarce polar environment. This directly aligns with the HESS scope of "physical, chemical, and biological processes within the hydrological cycle" and its emphasis on "the interaction of hydrology with other earth system sciences." Understanding these processes in Antarctica is vital for predicting freshwater availability for research stations, assessing the stability of ice shelves influenced by supraglacial lakes, and modeling regional climate feedbacks.

**R1. Thank you!**

2. Does the paper present novel concepts, ideas, tools, or data?

Yes, primarily through its novel data. The core novelty is the presentation of rare, direct eddy-covariance measurements of lake evaporation in coastal East Antarctica. This dataset is a significant contribution in itself. The development and validation of a wind dependent parameterization for the bulk transfer coefficient (C\_E) specifically for Antarctic lakes is a novel and valuable methodological outcome. While the concepts (EC, bulk method) are established, their application and rigorous validation in this extreme environment provide novel insights.

**R2. Thank you!**

3. Are substantial conclusions reached?

Yes. The conclusions are robust, significant, and well-supported by the data:

- Direct evaporation rates are quantified (0.3 to 5.0 mm d-1), showing clear dependence on ice cover and wind speed.
- Most combination formulas (Penman, Odrova, etc.) are shown to have severe systematic biases, underestimating evaporation by 27-73%.
- The bulk-aerodynamic method is confirmed to be highly accurate (6-8% bias) but only when using appropriate, site-specific transfer coefficients (e.g., from Arya (1988)), not generic ones.
- Wind speed is identified as the primary driver of short-term evaporation variability, a finding that contrasts with studies in less windy environments (like the Tibetan Plateau in Wang et al. (2019)).
  - Wang, B., Ma, Y., Ma, W., Su, B., & Dong, X. (2019). Evaluation of ten methods for estimating evaporation in a small high-elevation lake on the Tibetan Plateau. Theoretical and Applied Climatology, 136(3), 1033-1045.

R2.1: We added the following text after line 303: "Our results show that the wind speed is the primary driver for the short-term variation of evaporation, and it contradicts with the

results for the lakes in Tibetan Plateau (Wang et al., 2019) where weather is, however, less windy than in coastal Antarctica", and on line 569: "Wang, B., Ma, Y., Ma, W., Su, B., Dong, X.: Evaluation of ten methods for estimating evaporation in a small high-elevation lake on the Tibetan Plateau. Theoretical and Applied Climatology, 136(3), 1033-1045, 2019."

- The authors did not comment on the role of solar radiation which is the main driver of evaporation and needs to be discussed, even they did not directly measure it.
- R3. We added the following text after line 360: "The solar radiation is in the beginning of the causal chain of factors controlling ice and snow melt, lake water temperature and evaporation. It is explicitly included in the energy balance method which is among the other indirect methods applied to estimate the evaporation (Finch and Calver, 2008). In this study, however, we focused on indirect methods where the solar radiation is implicitly included in calculations because we did not measure the solar radiation in our experiments."
- 4. Are the scientific methods and assumptions valid and clearly outlined? Yes. The methods are state-of-the-art. The use of EC as a reference is the gold standard. The post-processing pipeline (spike removal, footprint filtering, gap-filling) is clearly described and follows established protocols. The assumptions (e.g., the applicability of Monin-Obukhov similarity theory, the representativeness of point measurements) are standard for such studies and are clearly addressed. The statistical analysis using RMSE and SSC is valid and appropriate.

**R4: Thank you!**

5. Are the results sufficient to support the interpretations and conclusions? Yes. The results are comprehensive and compelling. The data from two different lakes and two summer seasons provide a robust basis for analysis. The figures (timeseries, diurnal cycles, scatter plots) and tables (method comparison, skill scores) effectively present the evidence. The clear gradient of performance across the different methods strongly supports the conclusion that parameterization is key. The finding that wind speed correlates better with evaporation than the vapor pressure deficit is convincingly demonstrated.

**R5: Thank you!**

6. Is the description of experiments and calculations sufficiently complete to allow reproduction?

Yes. The description of the instrumentation, sensor heights, data processing steps, and equations is excellent. The provision of code and data on Zenodo is a major strength that ensures full reproducibility and aligns with best practices in open science.

**R6: Thank you!**

7. Do the authors give proper credit and indicate their original contribution? Yes. The introduction and discussion thoroughly contextualize the work within existing literature on polar hydrology and evaporation methods. The authors clearly reference the original sources of the combination formulas they test. Their own original contribution—the unique EC dataset and the

subsequent validation of methods—is clearly stated and forms the central pillar of the paper.

Note while Wang et al. (2019) focused on a different environment, a discussion
acknowledging that their finding (mass transfer methods work well) aligns with
conclusions from other extreme environments (like high-altitude lakes) could further
strengthen the context.

R7: We included the text after line 303and on line 363: "Our results show that the mass transfer methods work well enough to reproduce the evaporation over the lakes in the Schirmacher oasis, and this is aligns with outcomes from the studies focused on the evaporation over the high-altitude lakes of Tibetan Plateau (Wang et al., 2019)."

8. Does the title clearly reflect the contents of the paper?

Yes. The title is accurate, specific, and concise, correctly reflecting the location, subject, and process studied.

**R8: Thank you!**

9. Does the abstract provide a concise and complete summary?

Yes. The abstract perfectly summarizes the objectives, methods, key results (including quantitative findings), and the main conclusion and recommendation.

**R9: Thank you!**

10. Is the overall presentation well structured and clear?

Yes. The paper follows a standard and logical structure. The flow is easy to follow, and the argument is built progressively.

**R10: Thank you!**

11. Is the language fluent and precise?

Yes. The language is clear, formal, and scientific. While there are a few minor grammatical quirks (e.g., "containerizing" on p1), they do not hinder understanding. The manuscript is well-written.

**R11: Thank you! We have tried our best to improve the language.**

- 12. Are mathematical formulae, symbols, abbreviations, and units correctly defined and used? Yes. Formulas are presented clearly. Symbols are defined upon first use (e.g., in the bulk formula on p6). Units are used consistently throughout (mm d-1, ms-1, etc.).
  - *Note:* In Table 1, the column "Sum" has units "mm p-1". This should be clarified to "mm per [33-day] period" to avoid ambiguity.

**R12*: We corrected the text accordingly.**

- 13. Should any parts of the paper be clarified, reduced, combined, or eliminated?
  - Clarify: The distinction between "SSC" and "SSg" in the text and Table 4 should be made consistent.

**R13: SSg was the typo, and corrected in the revised version.**

• Clarify: The discussion of spray evaporation (p19) references "Eqs 3, 4", but only Eqs. 2 and 3 are presented. This should be corrected.

**R13: We corrected the text.**

• One needs to guess the applicability of Eqs 2,3, the meaning of the coefficients and the height where the wind speed w2 is measured should be explicitly stated.

**R13: The text was corrected.**

• L146: the formula for  $\sigma$  should appear before 'where' in L145.

**R13: Corrected.**

14. Are the number and quality of references appropriate?

Yes. The reference list is extensive, relevant, and includes key historical works, foundational methodological papers, and recent literature. It appropriately covers the fields of micrometeorology, Antarctic science, and hydrological methods.

**R14: Thank you!**

15. Is the amount and quality of supplementary material appropriate? Yes, and it is a significant strength. The availability of the raw code and data on Zenodo is exemplary and exceeds typical standards. It ensures full transparency and allows for exact reproduction of the analysis, which is crucial for a validation study like this.

**R15: Thank you!**

In the revised manuscript, we implemented the modifications following the comments of three reviewers. Also, besides of the modifications suggested by the reviewers, we implemented the following changes:

Fig. 8 became new Fig. 5 and its legend was modified.

Figure 5. The diurnal cycles of air temperature (a) and the lake surface water temperature (b) measured on Lake Zub/Priyadarshini (light blue) and Lake Glubokoe (coral).

The diurnal cycle of the air temperature is qualitatively similar for both experiments: it reaches the maximum at the mid-day hours and the minimum at midnight. The average temperature was, however, higher in December 2019 – January 2020 (Fig. 5 a, coral boxplots) than in January – February 2018 (Fig. 5 a, light blue boxplots). The diurnal cycle of the LSWT differs for the two lakes: on Lake Zub/Priyadarshini LWST shows the nighttime (23:00 –02:00) minimums of 3.0 °C and daytime (12:00–14:00) maximums up to 6.0 °C (Fig. 5 b, light blue boxplots), whereas on Lake Glubokoe, the LSWT showed a weaker diurnal cycle, LWST remaining close to 4.0 °C (Fig. 5 b, coral boxplots).

The Fig. 9 and its legend were modified as follows:

Figure 9. The diurnal cycle of the evaporation (a), wind speed (b) and saturation vapor pressure deficit (c) for two experiments on Lake Zub/Priyadarshini (coral) and Lake Glubokoe (light blue).

The text on lines 213-225 was modified as follows: "The diurnal cycle of evaporation over the lakes depends on the ice cover: the cycle is large during the ice-free stage on Lake Zub/Priyadarshini, and evaporation reaches its maximum (0.2 mm  $h^{-1}$ ) between 11:00 AM and 01:00 PM (Fig. 9 a, light blue boxes). Its diurnal cycle is similar to the cycle of saturation vapor pressure deficit (Fig. 9 c, light blue boxes). The strongest wind speed was observed at nighttime (03:00 – 04:00 AM) reaching up to 10 m s-1, while wind was often lower at daytime (04:00 – 05:00 PM). The diurnal cycle of evaporation over the partly ice-covered Lake Glubokoe showed maximum (0.15 mm  $h^{-1}$ ) in the early morning at 06:00–8:00 AM (Fig. 9 a, coral boxes), then reducing to even near-zero values in the late evening hours (09:00 – 11:00 PM) and night. It was in the opposite phase relative to the diurnal cycle of the wind speed, demonstrating that the saturation deficit dominated over wind speed as the primary driver of evaporation (Fig. 9). These different patterns in the diurnal cycle of

evaporation over the lakes reflect the complex interplay of factors (air-water temperature gradient, air humidity, wind speed, solar radiation) that vary throughout the day."

On lines 246 - 249 we added the text reads as follows: "We also calculated the transfer coefficient of moisture on the basis of the measurements on Lake Glubokoe following Andreas (1986), Arya (1988) and Fedorovich et al. (1991). The coefficient varied between  $1.46 \cdot 10^3$  and  $2.10 \cdot 10^3$  depending on the parameterization (Table 2). In Table 2, the transfer coefficients are presented for the measurement heights (different in the two experiments) and, to compare our results, also for the standard height of 10 meters ( $C_{EN10}$ )."

We modified the text on lines 319-320: "The evaporation over lakes is often evaluated applying combination formulas, and we found five combination formulas (Penman, 1948; Doorenbos and Pruitt, 1975; Odrova, 1979; Shuttleworth, 1993; and Shevnina et al., 2022) that have been applied over the lakes in Antarctica."

We added the text after line 362: "We did not present the estimations of the lake ice cover fraction from the digital images collected in the experiment on Lake Glubokoe, which may be a topic of the next study."

We added the text after line 349: "Lakes affect formation of fogs: passing of warm and moist air from lakes moves over colder ice covered surfaces cools the air to its dew point, leading to local fog (Gultepe et al., 2003) and precipitation (Su et al., 2020). In the Schirmacher oasis, the fog and "white rainbow" were observed in the early morning on 26 December 2020 (Fig. 12) when the relative humidity was over 95 %, the difference between the temperature of air and the lake water was close to –10 °C and wind speed was less than 1.0 ms-1. Such fogs may foster the surface melt, decrease visibility and make danger for the transport operations between the settlements, ice runways and coastal bases."

Fig. 12. The fog and "white rainbow" was observed on the early morning 26 December 2020 in the Schirmacher oasis (photo D. Emelyanov).

New references: "Gilson, G.F., Jiskoot, H., Cassano, J.J. *et al.*: The Thermodynamic Structure of Arctic Coastal Fog Occurring During the Melt Season over East Greenland. *Boundary-Layer Meteorol* 168, 443–467, https://doi.org/10.1007/s10546-018-0357-3, 2018

Su, D., Wen, L., Gao, X., Leppäranta, M., Song, X., Shi, Q., Kirillin, G.: Effects of the largest lake of the Tibetan plateau on the regional climate. J Geophys Res: Atmos 125, e2020JD033396, 2020."

We modified the text on lines 91-96: "This lake often stayed free of ice during 6–8 weeks during the austral summer (Khare et al., 2008), and it was ice free from 29–31 December 2017 to 8–12 February 2018 and from 22 – 25 December 2019 to 10 – 14 February 2020. Lake Glubokoe is of a maximum depth of 34.5 m (mean of depth is 13.1 m) and the surface area of 147000 m (Loopman et al., 1988). The lake is normally ice-covered year round (Kaup, 2005), but in recent years the lake has been ice free almost every summer (Sharov and Tolstikov, 2020). In February 2018 and 2020 the lake was ice free for two weeks."

We modified the text on lines 295 – 303: To our best knowledge, the direct (EC) observations on lake evaporation have been done in Antarctica for the lakes in the Schirmacher oasis. Ones, however, can be found for the regions with the polar desert climate, like the high-altitudes lakes located in the Qinghai-Tibet Plateau (36°N, 3194 m asl). Applying two years of observations, Li et al. (2016) found that the evaporation over the ice-free Lake Qinghai reaches up to 12 mm d-1 during wind storms, and days with wind speed stronger than 4 ms-1 contribute up 22 % to annual lake evaporation. Lately, Shi et al. (2024) found that the evaporation over the ice free lake is controlled

by wind based on the EC observations collected in 2014 – 2019. The authors suggested that the ice sublimation takes 23 % of annual evaporation over Lake Qinghai."

We changed the text on lines 362-370: "The method used in calculation of the evaporation is important for shallow coastal lakes whose volume changes very little over an austral summer (Shevnina and Kourzeneva, 2017). Gopinath et al., (2020) affirm the importance of the summertime evaporation over shallow lakes in the Schirmacher oasis applying the information on water isotope composition. Lake Zub/Priyadarshini is one of such shallow lakes whose water is used to supply the Maitri station and the new Maitri-II site (planned to open in 2029). The lake's water balance equation includes the precipitation, inlet river runoff (as inflow/positive component) and outlet river runoff, evaporation and water withdrawal (as outflow/negative component). The lake volume is 1032500 m³, and in January-February 2018 it decreased by 40.3 m³, and the discrepancies of the water balance equation was 670.6 m³ (Dhote et al., 2021). The lake evaporation of 58.5 m³ was calculated after Odrova (1979), which underestimated the evaporation for 72 % according to our results. In absolute values it corresponds to 42.1 m³ (58.5 multiplied by 0.72), which represents approximately 6 % of the discrepancy of the water balance equation. It can be reduced by using a better indirect method while calculating the evaporation."

We added the reference: Gopinath, G., Resmi, T. S., Praveenbabu, M., Pragath, M., Sunil, P. S., Rawat, S.: Isotope hydrochemistry of the lakes in Schirmacher Oasis, East Antarctica. Indian Journal of Geo Marine Sciences Vol. 49 (6), 947-953, 2020.

We also try our best to smooth the language of the overall narrative and prepare new supplements with the modified code used for plotting the figures.

Elena Shevnina from behalf of co-authors

---

## Author Comment (AC2)

**Response to Reviewer 2**

Our responses (R) are marked in red below each comment. In our revision, we implemented the following Specific Comments (SC):

SC1: Result section is still unclear for me and should be reorganized. I am still unclear whether the measuring time span the whole ice-free period of the two lakes. The author should also show the result of both lakes one by one, so the readers can be easy to follow.

RSC1: The eddy-covariance measurements were collected on two lakes in two different experiments in the years 2017 – 2018 on Lake Zub and in 2019 – 2020 on Lake Glubokoe. We measured evaporation during the ice-free period for the first lake (Zub), and it was planned that the measurements for the second lake (Glubokoe) will cover the whole summer (ice break-up and free periods). However, from mid-January 2020 our instrumentation was technically broken, and the measurements only covered the ice break-up period for Lake Zub. The results for the first experiment (Lake Zub) were published in Shevnina et al. (2022), however, they lack deriving site specific transfer coefficients for the bulk-aerodynamic method. This study fills the gap, and also targets (a) to estimate evaporation from the direct measurements collected in the second experiment (Lake Glubokoe), and (b) to test the empirical coefficients for the bulk-aerodynamic and combination formulas developed out of the measurements collected in the two experiments. We added new text after line 62: "Also, site-specific transfer coefficients for the bulk-aerodynamic method have not yet been suggested, but this study fills the gap."

SC2: Section 3.1, the authors should clarify whether it is the whole ice-free period or not for both lakes. If the EC system was only operated only part of the ice-free period, the authors should also clarify. For example, Figure 5 shows that the lake was still ice-free on Feb 25, 2020, but the evaporation result only covers the period from 7 December 2019 to 8 January 2020 (Line 183). The authors should give clarify this in the revision.

RSC2: The second experiment was planned to cover the whole summer period (ice-break-up and ice-free stages on Lake Glubokoe), but the logging system of IRGASON was broken in the middle of the season without possibility of being repaired in the field. We rewrote the text on lines 99–106 as follows: "The air temperature, barometric pressure, wind speed/direction and water vapour concentration were measured on a tower equipped with the EC open-path system IRGASON by Campbell Scientific. These measurements were collected on two lakes (Fig. 2 a) in two different experiments covering 38 days in 2018 and 33 days in 2019 – 2020. In the first experiment on Lake Zub/Priyadarshini, the evaporation was measured during the period from 1 January to 8 February 2018 when the lake was free of ice. The second experiment on Lake Glubokoe covered the period from 7 December 2019 to 8 January 2020. The experiment was planned to be carried out over the

duration of the austral summer, but our eddy-covariance instrumentation was damaged in mid-January. Hence, the actual measurements on Lake Glubokoe only represented its ice break-up period."

SC3: Line 101-103, lake water temperature should be addressed in the next paragraph. Please reorganize this section and the following.

RSC3: The text about the lake water temperature was moved to the next paragraph.

SC4: Section 4.2, The EC system was operated at both lakes. Evaporation over Lake Glubokoe was addressed, but the result of Lake Zub is not mentioned, why?

RSC4: This manuscript does not include the results for Lake Zub because they have been published previously in Shevnina et al. (2022). This previous paper, however, did not include (a) derivation of site-specific bulk-transfer coefficients and (b) performance of the empirical formulas evaluated against independent data. The present manuscript addressed these issues using the measurements collected on both lakes. All the above is clarified in the revised manuscript.

SC5: Figure 1, it is difficult to read. I can not find where are year round (red) and seasonal (blue) settlements from the Figure.

**RSC5: Figure 1 and its legend were modified as follows: "**

Fig. 1. Location of the Schirmacher oasis (SA) (a) and its infrastructure (b): year round (red dots) and seasonal (blue dots) settlements connected by roads (yellow lines, © Humanitarian OpenStreetMap Team (HOT), 2020. Distributed under the Open Data Commons Open Database License (ODbL) v1.0.). © Google Maps, 2019. The

red boxes in (a) and (b) outline the area with the main infrastructure in SA, and the image (c) shows the melted road to the White Desert Camp, December 2019 (photo D. Emelyanov)."

We replaced the text in lines 73-76 with the following text: "The oasis shelters two scientific bases operated year round since the 1960s (red dots within the red box in Fig. 1 b), and two tourist camps occupied seasonally (blue dots, Fig. 1 b). The scientific bases are occupied by 25–30 overwintering personnel, and up to 50 personnel during summer seasons. Up to 200 people can visit the tourist camps during summer. Two ice runways support transportation of people and cargo. Fuels are mostly delivered by ships to coastal bases located on the ice shelf, and then transported by vehicles to the settlements. The settlements, ice runways and coastal bases are connected with year-round ice roads (yellow lines in Fig. 1 b). In summertime, the transportation along the ice roads suffers from melted lakes and temporal streams formed over the ice surface. Figure 1 d shows the melted ice road to the White Desert Camp in December 2019."

SC6: Figure 4, what do the vertical dashed lines mean? The start and end of ice should be marked in the Figure

RSC6: We improved Figure 4 and its legend in lines 162 - 170 as follows:

Figure 4. The daily minimum, average and maximum for the air temperature (red lines) and LSWT (blue lines) measured on the shore of Lake Zub/Priyadarshini. The greed dashed lines show the beginning of the ice free period; the red lines on (b) show the dates of the lake images given in Fig. 6 b-d.

We added the following text on line 171: "During most of the days in both experiments, Lake Zub/Priyadarshinis and Lake Glubokoe were warmer than the ambient air. During the period from 30 December 2018 to 9 February 2019, the mean daily LSWT in Lake Zub/Priyadarshini was 3.9 °C, which was 4.7 °C higher than the mean air temperature (Fig. 4 a). The difference between the

LSWT and air temperature varied from -0.5 °C (2–3 January, 2018) to 10 °C (25–26 January, 2018). From 7 December 2019 to 15 February 2020, the mean daily LSWT of Lake Glubokoe was 3.1 °C, ranging from 0.6 to 5.3 °C. During this period, the lake was 2.4 °C warmer than the air on average (Fig. 4 b). The largest difference between the LWST and air temperature was observed on 25–26 December 2019, when also the relative humidity was the highest, exceeding 90 %."

SC7: Figure 5, The middle picture is taken from a different site compared with the other two.

Figure 6. The instrumentation installed on Lake Glubokoe (a), © Google Maps, 2019; (b, c and d) show the panoramic images of the lake location of the HOBO logger (red dot) and the EC system (yellow dot) on 3 January, 14 January and 25 February 2020 (photos D. Emelyanov)."

We added the on line 181: "On Lake Glubokoe, the ice break-up period lasted from 8-12 December until 12-15 February, and the ice-free period lasted for approximately two weeks. The lake-ice cover was documented in a series of digital images taken from the three positions marked as Camera 1, Camera 2 and Camera 3 in Fig. 6a. Fig. 6b and 6c show the examples of the lake images taken on 3 January 2020 and 25 February 2020 using Camera 3 in different stages of the ice cover: the ice break-up (b) and the ice free (c). Fig. 6d was taken 14 January 2020 from the position of Camera 2. The EC measurements were taken during those 33 days when the ice had melted from 30–35 % of the lake surface. The fraction of lake ice was evaluated by processing the digital images of the lake ice cover (taken every 5-10 days in the period of December 2019 – February 2020)."

In the revised manuscript, we implemented the modifications following the comments of three reviewers. Also, besides of the modifications suggested by the reviewers, we implemented the

**following changes:**

Fig. 8 became new Fig. 5 and its legend was modified.

Figure 5. The diurnal cycles of air temperature (a) and the lake surface water temperature (b) measured on Lake Zub/Priyadarshini (light blue) and Lake Glubokoe (coral).

The diurnal cycle of the air temperature is qualitatively similar for both experiments: it reaches the maximum at the mid-day hours and the minimum at midnight. The average temperature was, however, higher in December 2019 – January 2020 (Fig. 5 a, coral boxplots) than in January – February 2018 (Fig. 5 a, light blue boxplots). The diurnal cycle of the LSWT differs for two lakes: on Lake Zub/Priyadarshini LWST shows the nighttime (23:00–02:00) minimums of 3.0 °C and daytime (12:00–14:00) maximums up to 6.0 °C (Fig. 5 b, light blue boxplots), whereas on Lake Glubokoe, the LSWT showed a weaker diurnal cycle, LWST remaining close to 4.0 °C (Fig. 5 b, coral boxplots).

The Fig. 9 and its legend were modified as follows:

Figure 9. The diurnal cycle of the evaporation (a), wind speed (b) and saturation vapor pressure deficit (c) for two experiments on Lake Zub/Priyadarshini (coral) and Lake Glubokoe (light blue).

The text on lines 213-225 was modified as follows: "The diurnal cycle of evaporation over the lakes depends on the ice cover: the cycle is large during the ice-free stage on Lake Zub/Priyadarshini, and evaporation reaches its maximum (0.2 mm  $h^{-1}$ ) between 11:00 AM and 01:00 PM (Fig. 9 a, light blue boxes). Its diurnal cycle is similar to the cycle of saturation vapor pressure deficit (Fig. 9 c, light blue boxes). The strongest wind speed was observed at nighttime (03:00 – 04:00 AM) reaching up to 10 m s-1, while wind was often lower at daytime (04:00 – 05:00 PM). The diurnal cycle of evaporation over the partly ice-covered Lake Glubokoe showed maximum (0.15 mm  $h^{-1}$ ) in the early morning at 06:00–8:00 AM (Fig. 9 a, coral boxes), then reducing to even near-zero values in the late evening hours (09:00 – 11:00 PM) and night. It was in the opposite phase relative to the diurnal cycle of the wind speed, demonstrating that the saturation deficit dominated over wind speed as the primary driver of evaporation (Fig. 9). These different patterns in the diurnal cycle of

evaporation over the lakes reflect the complex interplay of factors (air-water temperature gradient, air humidity, wind speed, solar radiation) that vary throughout the day."

On lines 246 – 249 we added the text reads as follows: "We also calculated the transfer coefficient of moisture on the basis of the measurements on Lake Glubokoe following Andreas (1986), Arya (1988) and Fedorovich et al. (1991). The coefficient varied between  $1.46 \cdot 10^3$  and  $2.10 \cdot 10^3$  depending on the parameterization (Table 2). In Table 2, the transfer coefficients are presented for the measurement heights (different in the two experiments) and, to compare our results, also for the standard height of 10 meters ( $C_{EN10}$ )."

We modified the text on lines 319-320: "The evaporation over lakes is often evaluated applying combination formulas, and we found five combination formulas (Penman, 1948; Doorenbos and Pruitt, 1975; Odrova, 1979; Shuttleworth, 1993; and Shevnina et al., 2022) that have been applied over the lakes in Antarctica."

We added the text after line 362: "We did not present the estimations of the lake ice cover fraction from the digital images collected in the experiment on Lake Glubokoe, which may be a topic of the next study."

We added the text after line 349: "Lakes affect formation of fogs: passing of warm and moist air from lakes moves over colder ice covered surfaces cools the air to its dew point, leading to local fog (Gultepe et al., 2003) and precipitation (Su et al., 2020). In the Schirmacher oasis, the fog and "white rainbow" were observed in the early morning on 26 December 2020 (Fig. 12) when the relative humidity was over 95 %, the difference between the temperature of air and the lake water was close to –10 °C and wind speed was less than 1.0 ms-1. Fogs may foster the surface melt, decrease visibility and make danger for the transport operations between the settlements, ice runways and coastal bases."

Fig. 12. The fog and "white rainbow" was observed on the early morning 26 December 2020 in the Schirmacher oasis (photo D. Emelyanov).

New references: "Gilson, G.F., Jiskoot, H., Cassano, J.J. *et al.*: The Thermodynamic Structure of Arctic Coastal Fog Occurring During the Melt Season over East Greenland. *Boundary-Layer Meteorol* 168, 443–467, <a href="https://doi.org/10.1007/s10546-018-0357-3">https://doi.org/10.1007/s10546-018-0357-3</a>, 2018

Su, D., Wen, L., Gao, X., Leppäranta, M., Song, X., Shi, Q., Kirillin, G.: Effects of the largest lake of the Tibetan plateau on the regional climate. J Geophys Res: Atmos 125, e2020JD033396, 2020."

We modified the text on lines 91-96: "This lake often stayed free of ice during 6–8 weeks during the austral summer (Khare et al., 2008), and it was ice free from 29–31 December 2017 to 8–12 February 2018 and from 22 – 25 December 2019 to 10 – 14 February 2020. Lake Glubokoe is of a maximum depth of 34.5 m (mean of depth is 13.1 m) and the surface area of 147000 m (Loopman et al., 1988). The lake is normally ice-covered year round (Kaup, 2005), but in recent years the lake has been ice free almost every summer (Sharov and Tolstikov, 2020). In February 2018 and 2020 the lake was ice free for 2 – 3 weeks."

We modified the text on lines 295 – 303: To our best knowledge, the direct (EC) observations on lake evaporation have been done in Antarctica for the lakes in the Schirmacher oasis. Ones, however, can be found for the regions with the polar desert climate, like the high-altitudes lakes located in the Qinghai-Tibet Plateau (36°N, 3194 m asl). Applying two years of observations, Li et al. (2016) found that the evaporation over the ice-free Lake Qinghai reaches up to 12 mm d-1 during wind storms, and days with wind speed stronger than 4 ms-1 contribute up 22 % to annual lake evaporation. Lately, Shi et al. (2024) found that the evaporation over the ice free lake is

controlled by wind based on the EC observations collected in 2014 – 2019. The authors suggested that the ice sublimation takes 23 % of annual evaporation over Lake Qinghai."

We changed the text on lines 362-370: "The method used in calculation of the evaporation is important for shallow coastal lakes whose volume changes very little over an austral summer (Shevnina and Kourzeneva, 2017). Gopinath et al., (2020) affirmed the importance of the summertime evaporation from shallow lakes in the Schirmacher oasis applying the information on water isotope composition. Lake Zub/Priyadarshini is one of such shallow lakes whose water is used to supply the Maitri station and the new Maitri-II site (planned to open in 2029). The lake's water balance equation includes the precipitation, inlet river runoff (as inflow/positive component) and outlet river runoff, evaporation and water withdrawal (as outflow/negative component). The lake volume is 1032500 m³, and in January-February 2018 it decreased by 40.3 m³, and the discrepancies of the water balance equation was 670.6 m³ (Dhote et al., 2021). The lake evaporation of 58.5 m³ was calculated after Odrova (1979), which underestimated the evaporation for 72 % according to our results. In absolute values it corresponds to 42.1 m³ (58.5 multiplied by 0.72), which represents approximately 6 % of the discrepancy of the water balance equation. It can be reduced by using a better indirect method while calculating the evaporation."

We added the reference: Gopinath, G., Resmi, T. S., Praveenbabu, M., Pragath, M., Sunil, P. S., Rawat, S.: Isotope hydrochemistry of the lakes in Schirmacher Oasis, East Antarctica. Indian Journal of Geo Marine Sciences Vol. 49 (6), 947-953, 2020.

We also try our best to smooth the language of the overall narrative and prepare new supplements with the modified code used for plotting the figures.

Elena Shevnina from behalf of co-authors

---

## Author Comment (AC3)

**Response to Reviewer 3**

Our responses (R) are marked in red below each comment. The manuscript got the revision by implementing the following Specific Comments (SC):

SC1: Listing the four combination formula simply may improve understanding of the precision of the used 6 methods.

RSC1: We added the following text on line 129: "To calculate the daily evaporation over Lake Glubokoe, we applied the combination equations, which read as follows:  $E = 0.26(1 + 0.54 w_2)(e_s - e_2)$  based on Penman (1948) and Tanny et al. (2008),  $E = 0.26(1 + 0.86 w_2)(e_s - e_2)$  according to Doorenbos and Pruitt (1975), and  $E = 2.909A^{-0.05}w_2(e_s - e_2)$  according to Shuttleworth (1993). We also used the formula  $E = 0.14(1 + 0.72w_2)(e_s - e_2)$  suggested for evaporation over lakes in northern Russia (Odrova 1979), and  $E = -0.33(1 - 1.82w_2)(e_s - e_2)$  suggested based on the direct measurements over Lake Zub/Priyadarshini (Shevnina et al. 2022). In these equations, E is the evaporation in mm day-1, E is the surface area in E is the 2 m wind speed in m s-1, es is the saturation water vapour pressure for the lake surface temperature, and E is the air water vapor pressure at the height of 2 m. Both es and E are in hPa, and calculated according to the Tetens's formula given in Stull (2017)."

SC2: In Section 3.1, it may be better to introduce all of the information one lake by one lake.

RSC2: We modified the text on lines 99–106 to read as follows: "The air temperature, atmospheric pressure, wind speed/direction, and water vapour concentration were measured on a tower equipped with the EC open-path system IRGASON by Campbell Scientific. These measurements were collected on two lakes (Fig. 2a) in two different experiments covering 38 days in 2018 and 33 days in 2019 – 2020. In the first experiment on Lake Zub/Priyadarshini, the evaporation was measured during the period from 1 January to 8 February 2018 when the lake was free of ice. The second experiment on Lake Glubokoe covers the period from 7 December 2019 to 8 January 2020. The experiment was planned to be carried out over the duration of the austral summer, but our eddy-covariance instrumentation was damaged in mid-January. Hence, the actual measurements on Lake Glubokoe only represented its ice break-up period."

SC3: The origin of the wind-dependent transfer coefficient (Eqs. 2-3, Fig. 10) is unclear. This is a critical part of the analysis and must be explicitly stated: are these relationships developed

from the Zub 2017-2018 data in this manuscript, or are they presented as established relationships from a previous study? This has major implications for interpreting the validation results in Table 3 and Fig. 11.

RSC3: The text on lines 136-138 was modified to read as follows: "The transfer coefficient of moisture was calculated following the parametrization schemes suggested by Heikinheimo et al. (1999), Andreas (1986), Arya (1988) and Fedorovich et al. (1991). The new wind-dependent relationship was derived by applying Equation (1) on the basis of data on evaporation, wind speed, air specific humidity and surface saturation specific humidity from Lake Zub during 2017-2018."

C4: If the CEin equation (2) and (3) were obtained in the previous work (Shevnina et al., 2022), it may appear in Section 3.2 Methods. Table 2 may be moved to the section, too.

RSC4: As explained above, the equations were derived in this study. We also modified the text on lines 57 – 59: "The bulk-aerodynamic method is often used to assess evaporation/sublimation over lakes and glaciers in Antarctica (Clow et al., 1988; Bliss et al., 2011; Leppäranta et al., 2016). In this method, the turbulent exchange (mass-transfer) coefficients account for atmospheric stability, which is calculated through Monin–Obukhov framework, incorporating empirical dimensionless gradient functions (Brutsaert, 1982). The empirical gradient functions are site specific (Guseva et al., 2023), and they can be evaluated from the eddy-covariance (EC) measurements on lakes (Franz et al., 2018; Ala-Könni et al., 2022). These gradient functions, however, have not yet been suggested for lakes in coastal Antarctica. The bulk aerodynamic method where the empirical gradient functions estimated for a boreal lake site underestimated the summertime evaporation over an ice-free lake by over 32% in Antarctica, and it is not clear how good it is for the lakes during the ice break-up period (Shevnina et al., 2022)."

SC5: Is Figure 10 from the above mentioned previous work? If not, how to get it?
RSC5. Fig. 10 is a part of this work. We modified the figure legend on line 245 as follows:
Figure 10. Dependence of the 10-m neutral transfer coefficient for moisture (CEN10) on the 2-m wind speed (w2, ms-1) over Lake Zub/Priyadarshini in 2017-2018.

SC6. The performance of the wind-dependent method needs explanation. If it was derived from Zub data, why does it not perform best for Zub? Why does the best parameterization

differ between the two lakes? This warrants discussion on the site specificity of these coefficients.

RSC6. We added explanations after line 283: "It is indeed interesting that the new wind-dependent transfer coefficient, derived on the basis of Lake Zub/Priyadarshini data, has a smaller RMSE for Lake Glubokoe than for Lake Zub/Priyadarshini. This is, however, understandable because the transfer coefficient is only one of the factors that controls the evaporation calculated using the bulk-aerodynamic formula (Eq. (1)). In addition to the transfer coefficient, at least the following factors may generate inaccuracy in the estimated evaporation: (1) errors in the measurements of lake surface temperature (controlling the saturation specific humidity), air specific humidity, and wind speed, (2) contribution of spray droplets to evaporation, (3) role of waves in the lake surface, and (4) validity of the Monin-Obukhov similarity theory, which requires quasi-stationary, horizontally homogeneous conditions. Errors and uncertainties associated with (1) - (4) above may either increase the overall error or partially offset one another. Hence, it is reasonable that the performance of parameterization schemes varies among lakes, as factors (1) - (4) may be site-specific, and also depends on the metrics used (e.g., RMSE or SSC) to evaluate performance (Table 4).

SC7: Figures: Most of subfigures and legends are unclear. It is recommended that each subfigure of a figure be labeled as a, b, c, d, etc., instead of distinguishing subfigures by terms like "Figure 2 top" or "Figure 2 bottom". Additionally, each piece of information in the figures should be explained in the legends.

RSC7: We modified the legends of Figures 1, 2, 3, 4, 5, 6, 7, 8, 9 and 10.

SC8: The format of the numbering is inconsistent with that of other figures (e.g., Figure 2). It is suggested to unify the figure numbering format, such as using "Figure 1" consistently with "Figure 2".

RSC8: We unified the numbering format for all figures/subfigures in the revised manuscript.

SC9: Section 4.1: The descriptions in the text do not correspond to the figures, leading to confusion. It is suggested that the authors carefully check and revise this part.

RSC9. We corrected the text on line 152 as follows: "During the experiment on Lake Zub/Priyadarshini, the weather was colder and less windy than the climatology over 1961 – 2010, estimated according to the observations at Novo meteorological site (Shevnina et al.,

2022). The daily air temperatures ranged between –8.3 and 2.8 °C. The wind speed varied between 1.5 to 14.3 ms-1, with the mean of 6.2 m s-1. The mean relative humidity was 54 %."

SC10. Figure 3: I only observed black and green colors in the upper subfigure. Where does "water vapour concentration (blue)" in the legend come from? Does "relative humidity" correspond to green or black? Are the data in the figure daily data or half hourly data? I assume they are half-hourly data, which would contradict the description in L152 that the range of daily air temperature is -4.9 to 5.1 °C. 11.

**RSC10. We corrected Figure 3 and its legend on lines 159 – 160 as follows:**

Figure 3: The mean (solid line), minimum and maximum (dashed lines) values for the air temperature (a), the relative humidity (b) and the wind speed (c) for the period of two experiments (y-axis is given in days starting from 1 of December). In legends: LG is Lake Glubokoe and ZB is Lake Zub/Priyadarshini.

SC1: L196-203: This section is analyzed based on Figure 7, yet there is no reference to Figure 7 at all. It is suggested that the authors add "(Fig. 7a)" and other corresponding references at appropriate places in the text.

**RSC11. Figure 7 and its legend were modified on lines 194 – 203:**

Figure 8. Time series of the 30-minute means of the wind direction (a) and evaporation (b) observed by the EC system on Lake Glubokoe. On (a): the green lines show the wind directions of 90 and 225 degrees (the footprint); on (b): the red dots indicate the measurements collected outside the footprint and the blue line shows the mean evaporation.

We added the text: "The raw 30-minute measurements were filtered by (a) the sensors' signal strengths, (b) number of gaps in the observations, (c) the footprint, and (d) the sector of wind directions covering the lake (the green lines in Fig. 8a). The percentage of the filtered out (excluded) measurements did not exceed 20 % of total data (red dots in Fig. 8b);most of the excluded data corresponded to wind directions outside the footprint. These gaps were replaced by the mean of evaporation estimated for the period of the experiment (the blue line in Fig. 8b). Since most of the gaps are lower than the mean, we also replaced them by the 25 percentile evaporation (the orange line on Fig. 8b). The evaporation over each day was calculated from 30 minute measurements (with the gaps filled), and the total sum for the period of the experiment is

50 and 54 mm (per period of 33 days) being replaced by the mean and 25 percentile evaporation, respectively. The daily mean evaporation was  $1.5 \pm 0.1$  mm d-1 and  $1.6 \pm 0.1$  mm d-1, and varied between 0.3 and 3.2 mm d-1. These estimates show that filling the gaps either by the mean or 25 percentile evaporation gave almost the same results, and we further filled them by the mean evaporation. The largest evaporation (more than  $2.5 \text{ mm d}^{-1}$ ) was observed on 9 - 11 December 2019 and 3 - 4 January 2020; and, the lowest evaporation (less than  $0.9 \text{ mm d}^{-1}$ ) was observed on 6, 7 January 2020.

In the revised manuscript, we implemented the modifications following the comments of three reviewers. Also, we implemented the following changes:

Figure 5. The diurnal cycles of air temperature (a) and the lake surface water temperature (b) measured on Lake Zub/Priyadarshini (light blue) and Lake Glubokoe (coral).

The diurnal cycle of the air temperature is qualitatively similar for both experiments: it reaches the maximum at the mid-day hours and the minimum at midnight. The average temperature was, however, higher in December 2019 – January 2020 (Fig. 5 a, coral boxplots) than in January – February 2018 (Fig. 5 a, light blue boxplots). The diurnal cycle of the LSWT differs for two lakes: on Lake Zub/Priyadarshini LWST shows the nighttime (23:00–02:00) minimums of 3.0 °C and

daytime (12:00–14:00) maximums up to 6.0 °C (Fig. 5 b, light blue boxplots), whereas on Lake Glubokoe, the LSWT showed a weaker diurnal cycle, LWST remaining close to 4.0 °C (Fig. 5 b, coral boxplots).

The Fig. 9 and its legend were modified as follows:

Figure 9. The diurnal cycle of the evaporation (a), wind speed (b) and saturation vapor pressure deficit (c) for two experiments on Lake Zub/Priyadarshini (coral) and Lake Glubokoe (light blue).

The text on lines 213-225 was modified as follows: "The diurnal cycle of evaporation over the lakes depends on the ice cover: the cycle is large during the ice-free stage on Lake Zub/Priyadarshini, and evaporation reaches its maximum (0.2 mm  $h^{-1}$ ) between 11:00 AM and 01:00 PM (Fig. 9 a, light blue boxes). Its diurnal cycle is similar to the cycle of saturation vapor pressure deficit (Fig. 9 c, light blue boxes). The strongest wind speed was observed at nighttime (03:00 – 04:00 AM) reaching up to 10 m s-1, while wind was often lower at daytime (04:00 – 05:00 PM). The diurnal cycle of evaporation over the partly ice-covered Lake Glubokoe showed maximum (0.15 mm  $h^{-1}$ ) in the

early morning at 06:00-8:00 AM (Fig. 9 a, coral boxes), then reducing to even near-zero values in the late evening hours (09:00-11:00 PM) and night. It was in the opposite phase relative to the diurnal cycle of the wind speed, demonstrating that the saturation deficit dominated over wind speed as the primary driver of evaporation (Fig. 9). These different patterns in the diurnal cycle of evaporation over the lakes reflect the complex interplay of factors (air-water temperature gradient, air humidity, wind speed, solar radiation) that vary throughout the day."

On lines 246 – 249 we added the text reads as follows: "We also calculated the transfer coefficient of moisture on the basis of the measurements on Lake Glubokoe following Andreas (1986), Arya (1988) and Fedorovich et al. (1991). The coefficient varied between  $1.46 \cdot 10^3$  and  $2.10 \cdot 10^3$  depending on the parameterization (Table 2). In Table 2, the transfer coefficients are presented for the measurement heights (different in the two experiments) and, to compare our results, also for the standard height of 10 meters ( $C_{EN10}$ )."

We modified the text on lines 319-320: "The evaporation over lakes is often evaluated applying combination formulas, and we found five combination formulas (Penman, 1948; Doorenbos and Pruitt, 1975; Odrova, 1979; Shuttleworth, 1993; and Shevnina et al., 2022) that have been applied over the lakes in Antarctica."

We added the text after line 362: "We did not present the estimations of the lake ice cover fraction from the digital images collected in the experiment on Lake Glubokoe, which may be a topic of the next study."

We added the text after line 349: "Lakes affect formation of fogs: passing of warm and moist air from lakes moves over colder ice covered surfaces cools the air to its dew point, leading to local fog (Gultepe et al., 2003) and precipitation (Su et al., 2020). In the Schirmacher oasis, the fog and "white rainbow" were observed in the early morning on 26 December 2020 (Fig. 12) when the relative humidity was over 95 %, the difference between the temperature of air and the lake water was close to –10 °C and wind speed was less than 1.0 ms-1. Fogs may foster the surface melt, decrease visibility and make danger for the transport operations between the settlements, ice runways and coastal bases."

Fig. 12. The fog and "white rainbow" was observed on the early morning 26 December 2020 in the Schirmacher oasis (photo D. Emelyanov).

New references: "Gilson, G.F., Jiskoot, H., Cassano, J.J. *et al.*: The Thermodynamic Structure of Arctic Coastal Fog Occurring During the Melt Season over East Greenland. *Boundary-Layer Meteorol* 168, 443–467, <a href="https://doi.org/10.1007/s10546-018-0357-3">https://doi.org/10.1007/s10546-018-0357-3</a>, 2018

Su, D., Wen, L., Gao, X., Leppäranta, M., Song, X., Shi, Q., Kirillin, G.: Effects of the largest lake of the Tibetan plateau on the regional climate. J Geophys Res: Atmos 125, e2020JD033396, 2020."

We modified the text on lines 91-96: "This lake often stayed free of ice during 6–8 weeks during the austral summer (Khare et al., 2008), and it was ice free from 29–31 December 2017 to 8–12 February 2018 and from 22 – 25 December 2019 to 10 – 14 February 2020. Lake Glubokoe is of a maximum depth of 34.5 m (mean of depth is 13.1 m) and the surface area of 147000 m (Loopman et al., 1988). The lake is normally ice-covered year round (Kaup, 2005), but in recent years the lake has been ice free almost every summer (Sharov and Tolstikov, 2020). In February 2018 and 2020 the lake was ice free for 2 – 3 weeks."

We modified the text on lines 295 – 303: To our best knowledge, the direct (EC) observations on lake evaporation have been done in Antarctica for the lakes in the Schirmacher oasis. Ones, however, can be found for the regions with the polar desert climate, like the high-altitudes lakes located in the Qinghai-Tibet Plateau (36°N, 3194 m asl). Applying two years of observations, Li et al. (2016) found that the evaporation over the ice-free Lake Qinghai reaches up to 12 mm d-1 during wind storms, and days with wind speed stronger than 4 ms-1 contribute up 22 % to annual lake evaporation. Lately, Shi et al. (2024) found that the evaporation over the ice free lake is

controlled by wind based on the EC observations collected in 2014 – 2019. The authors suggested that the ice sublimation takes 23 % of annual evaporation over Lake Qinghai."

We changed the text on lines 362-370: "The method used in calculation of the evaporation is important for shallow coastal lakes whose volume changes very little over an austral summer (Shevnina and Kourzeneva, 2017). Gopinath et al., (2020) affirmed the importance of the summertime evaporation from shallow lakes in the Schirmacher oasis applying the information on water isotope composition. Lake Zub/Priyadarshini is one of such shallow lakes whose water is used to supply the Maitri station and the new Maitri-II site (planned to open in 2029). The lake's water balance equation includes the precipitation, inlet river runoff (as inflow/positive component) and outlet river runoff, evaporation and water withdrawal (as outflow/negative component). The lake volume is 1032500 m³, and in January-February 2018 it decreased by 40.3 m³, and the discrepancies of the water balance equation was 670.6 m³ (Dhote et al., 2021). The lake evaporation of 58.5 m³ was calculated after Odrova (1979), which underestimated the evaporation for 72 % according to our results. In absolute values it corresponds to 42.1 m³ (58.5 multiplied by 0.72), which represents approximately 6 % of the discrepancy of the water balance equation. It can be reduced by using a better indirect method while calculating the evaporation."

We added the reference: Gopinath, G., Resmi, T. S., Praveenbabu, M., Pragath, M., Sunil, P. S., Rawat, S.: Isotope hydrochemistry of the lakes in Schirmacher Oasis, East Antarctica. Indian Journal of Geo Marine Sciences Vol. 49 (6), 947-953, 2020.

We also try our best to smooth the language of the overall narrative and prepare new supplements with the modified code used for plotting the figures.

Elena Shevnina from behalf of co-authors